# Functional control of oscillator networks

Tommaso Menara [1], Giacomo Baggio[2], Dani Bassett [3,4] & Fabio Pasqualetti [5✉]

Oscillatory activity is ubiquitous in natural and engineered network systems. The interaction scheme underlying interdependent oscillatory components governs the emergence of network-wide patterns of synchrony that regulate and enable complex functions. Yet, understanding, and ultimately harnessing, the structure-function relationship in oscillator networks remains an outstanding challenge of modern science. Here, we address this challenge by presenting a principled method to prescribe exact and robust functional configurations from local network interactions through optimal tuning of the oscillators' parameters. To quantify the behavioral synchrony between coupled oscillators, we introduce the notion of *functional pattern*, which encodes the pairwise relationships between the oscillators' phases. Our procedure is computationally efficient and provably correct, accounts for constrained interaction types, and allows to concurrently assign multiple desired functional patterns. Further, we derive algebraic and graph-theoretic conditions to guarantee the feasibility and stability of target functional patterns. These conditions provide an interpretable mapping between the structural constraints and their functional implications in oscillator networks. As a proof of concept, we apply the proposed method to replicate empirically recorded functional relationships from cortical oscillations in a human brain, and to redistribute the active power flow in different models of electrical grids.

[1] Department of Mechanical and Aerospace Engineering, University of California, San Diego, La Jolla, CA 92093, USA. [2] Department of Information Engineering, University of Padova, Padova 35131, Italy. [3] Department of Physics & Astronomy, Department of Bioengineering, Department of Electrical & Systems Engineering, Department of Neurology, Department of Psychiatry, University of Pennsylvania, Philadelphia, PA 19104, USA. [4] The Santa Fe Institute, Santa Fe, NM 87506, USA. [5] Department of Mechanical Engineering, University of California, Riverside, Riverside, CA 92521, USA. ✉email: fabiopas@engr.ucr.edu

The complex coordinated behavior of oscillatory components is linked to the function of many natural and technological network systems[1–3]. For instance, distinctive network-wide patterns of synchrony[4–6] determine the coordinated motion of orbiting particle systems[7], promote successful mating in populations of fireflies[8], regulate the active power flow in electrical grids[9], predict global climate change phenomena[10], dictate the structural development of mother-of-pearl in mollusks[11], and enable numerous cognitive functions in the brain[12,13]. Since this rich repertoire of patterns emerges from the properties of the underlying interaction network[14], controlling the collective configuration of interdependent units holds tremendous potential across science and engineering[15]. Despite its practical significance, a comprehensive method to enforce network-wide patterns of synchrony by intervening in the network's structural parameters does not yet exist.

In this work, we develop a rigorous framework that allows us to optimally control the spatial organization of the network components and their oscillation frequencies to achieve desired patterns of synchrony. We abstract the rhythmic activity of a system as the output of a network of diffusively coupled oscillators[16,17] with Kuramoto dynamics. This modeling choice is motivated by the rich dynamical repertoire and wide adoption of Kuramoto oscillators[18]. Specifically, we consider an undirected network $\mathcal{G} = \{\mathcal{O}, \mathcal{E}\}$ of $n$ oscillators with dynamics

$$\dot{\theta}_i = \omega_i + \sum_{j=1}^{n} A_{ij} \sin(\theta_j - \theta_i), \qquad (1)$$

where $\omega_i \in \mathbb{R}$ and $\theta_i \in \mathbb{S}^1$ are the frequency and phase of the $i$th oscillator, respectively, $A = [A_{ij}]$ is the weighted adjacency matrix of $\mathcal{G}$, and $\mathcal{O} = \{1, \dots, n\}$ and $\mathcal{E} \subseteq \mathcal{O} \times \mathcal{O}$ denote the oscillator and interconnection sets, respectively. In this work, we consider the case where the network $\mathcal{G}$ admits both cooperative (i.e., $A_{ij} > 0$) and competitive (i.e., $A_{ij} < 0$)[19] interactions among the oscillators, as well as the more constrained case of purely cooperative interactions that arises in several real-word systems. For instance, negative interactions are not physically meaningful in networks of excitatory neurons, in power distribution networks (where the interconnection weight denotes conductance and susceptance of a transmission line), and in urban transportation

networks (where interconnections denote the number of vehicles on a road with respect to its maximum capacity).

To quantify the pairwise functional relations between oscillatory units, and inspired by the work in ref. [20], we define a local order parameter that, compared to the classical Pearson correlation coefficient, does not depend on sampling time and is more convenient when dealing with periodic phase signals (see Supplementary Information). Given a pair of phase oscillators $i$ and $j$ with phase trajectories $\theta_i(t)$ and $\theta_j(t)$, we define the correlation coefficient

$$\rho_{ij} = \langle \cos(\theta_j(t) - \theta_i(t)) \rangle_t, \qquad (2)$$

where $\langle \cdot \rangle_t$ denotes the average over time. A *functional pattern* is formally defined as the symmetric matrix $R$ whose $i, j$th entry equals $\rho_{ij}$. Importantly, a functional pattern explicitly encodes the pairwise, local, correlations across all of the oscillators, which are more informative than a global observable (e.g., the order parameter[16,21]). It is easy to see that, if two oscillators $i$ and $j$ synchronize after a certain initial transient, $\rho_{ij}$ converges to 1 as time increases. If two oscillators $i$ and $j$ become phase-locked (i.e., their phase difference remains constant over time), then their correlation coefficient converges to some constant value with a magnitude smaller than 1. If the phases of two oscillators $i$ and $j$ evolve independently, then their correlation value remains small over time. A few questions arise naturally, which will be answered in this paper. Are all functional patterns achievable? Which network configurations allow for the emergence of multiple target functional patterns? And, if a certain functional pattern can be achieved, is it robust to perturbations? Surprisingly, we reveal that controlling functional patterns can be cast as a convex optimization problem, whose solution can be characterized explicitly. Figure 1 shows our framework and an example of control of functional patterns for a network with 7 oscillators. In the paper, we will validate our methods by replicating functional patterns from brain recordings in an empirically reconstructed neuronal network, and by controlling the active power distribution in multiple models of the power grid.

While synchronization phenomena in oscillator networks have been studied extensively (e.g., see refs. [22–26]), the development of control methods to impose desired synchronous behaviors has

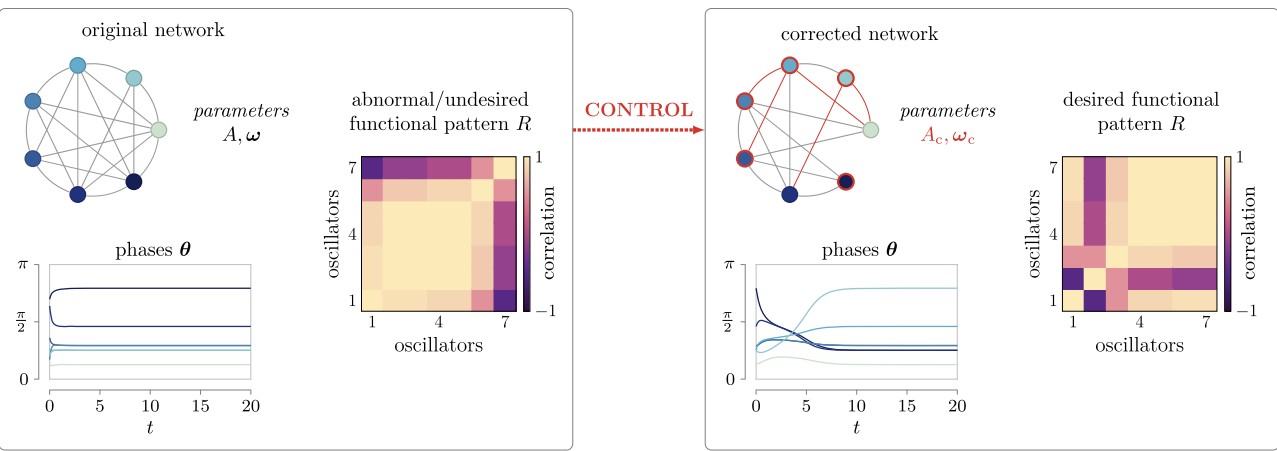

**Fig. 1 Network control to enforce a desired functional pattern from an abnormal or undesired one.** The left panel contains a network of $n = 7$ oscillators (top left panel, line thickness is proportional to the coupling strength), whose vector of natural frequencies $\boldsymbol{\omega}$ has zero mean. The phase differences with respect to $\theta_1$ (i.e., $\theta_i - \theta_1$) converge to $\left[\frac{\pi}{8} \frac{\pi}{8} \frac{\pi}{6} \frac{\pi}{6} \frac{\pi}{6} \frac{2\pi}{3}\right]$, as also illustrated in the phases' evolution from random initial conditions (bottom left panel, color coded). The center left panel depicts the functional pattern $R$ corresponding to such phase differences over time. The right panel illustrates the same oscillator network after a selection of coupling strengths and natural frequencies have been tuned (in red, the structural parameters $A$ and $\boldsymbol{\omega}$ are adjusted to $A_c$ and $\boldsymbol{\omega}_c$) to obtain the phase differences $\left[\frac{2\pi}{3} \frac{\pi}{3} \frac{\pi}{6} \frac{\pi}{6} \frac{\pi}{8} \frac{\pi}{8}\right]$, which encode the desired functional pattern in the center right. In this example, we have computed the closest set (in the $\ell_1$-norm sense) of coupling strengths and natural frequencies to the original ones that enforce the emergence of the target pattern. Importantly, only a subset of the original parameters has been modified.

only recently attracted the attention of the research community[27–30]. Perhaps the work that is closest to our approach is ref. [28], where the authors tailor interconnection weights and natural frequencies to achieve a specified level of phase cohesiveness in a network of Kuramoto oscillators. Our work improves considerably upon this latter study, whose goal is limited to prescribing an upper bound to the phase differences, by enabling the prescription of pairwise differences and by investigating the stability properties of different functional patterns. Taken together, existing results highlight the importance of controlling distinct configurations of synchrony, but remain mainly focused on the control of "macroscopic" observables of synchrony (e.g., the average synchronization level of all the oscillators). In contrast, our control method prescribes desired pairwise levels of correlation across all of the oscillators, thus enabling a precise "microscopic" description of functional interactions.

## Results

**Feasible functional patterns in positive networks.** A functional pattern is an $n \times n$ matrix whose entries are the time-averaged cosine of the differences in the oscillator phases (see Eq. (2)). When the oscillators reach an equilibrium, the differences of the oscillator phases become constant, and the network evolves into a *phase-locked* configuration. In this case, the functional pattern of the network also becomes constant and is uniquely determined by the phase differences at the equilibrium configuration. In this work, we study functional patterns for the special case of phase-locked oscillators and, since a functional pattern can be specified using a set of phase differences at equilibrium, convert the problem of generating a functional pattern into the problem of ensuring a desired phase-locked equilibrium. We recall that, while convenient for the analysis, phase-locked configurations play a crucial role in the functioning of many natural and man-made networks[31–33].

For the undirected network $\mathcal{G} = (\mathcal{O}, \mathcal{E})$, let $x_{ij} = \theta_j - \theta_i$ be the difference of the phases of the oscillators $i$ and $j$, and let $\boldsymbol{x} \in \mathbb{R}^{|\mathcal{E}|}$ be the vector of all phase differences with $(i, j) \in \mathcal{E}$ and $i < j$. The network dynamics (1) can be conveniently rewritten in vector form as (see the "Methods" section)

$$BD(\boldsymbol{x})\boldsymbol{\delta} = [\omega_1 \cdots \omega_n]^\mathsf{T} - \dot{\boldsymbol{\theta}}, \qquad (3)$$

where $B \in \mathbb{R}^{n \times |\mathcal{E}|}$ is the (oriented) incidence matrix of the network $\mathcal{G}$, $D(\boldsymbol{x}) \in \mathbb{R}^{|\mathcal{E}| \times |\mathcal{E}|}$ is a diagonal matrix of the sine functions in Eq. (1), and $\boldsymbol{\delta} \in \mathbb{R}^{|\mathcal{E}|}$ is a vector collecting all the weights $A_{ij}$ with $i < j$. Because we focus on phase-locked trajectories, all oscillators evolve with the same frequency and the vector $\dot{\boldsymbol{\theta}}$ satisfies $\dot{\boldsymbol{\theta}} = \omega_{\mathrm{mean}} \mathbf{1}$, where $\omega_{\mathrm{mean}} = \left(\frac{1}{n} \sum_{i=1}^n \omega_i\right)$ is the average of the natural frequencies of the oscillators. Further, since $\mathcal{G}$ contains only $n$ oscillators, any phase difference $x_{ij}$ can always be written as a function of $n-1$ independent differences; for instance, $\{x_{12}, x_{23}, \ldots, x_{n-1,n}\}$. For instance, for any pair of oscillators $i$ and $j$ with $i < j$, it holds $x_{ij} = \sum_{k=i}^{j-1} x_{k,k+1}$. This implies that the vector of all phase differences in equation (3), and in fact any $n \times n$ functional pattern, has only $n - 1$ degrees of freedom and can be uniquely specified with a set of $n - 1$ independent differences $\boldsymbol{x}_{\mathrm{desired}}$ (see the "Methods" section). Following this reasoning and to avoid cluttered notation, let $\boldsymbol{\omega} = [\omega_1 - \omega_{\mathrm{mean}} \cdots \omega_n - \omega_{\mathrm{mean}}]^\mathsf{T}$, and notice that the problem of enforcing a desired functional pattern simplifies to (i) converting the desired functional pattern to the corresponding phase differences $\boldsymbol{x}_{\mathrm{desired}}$, and (ii) computing the network weights $\boldsymbol{\delta}$ to

satisfy the following equation:

$$BD(\boldsymbol{x})\boldsymbol{\delta} = \boldsymbol{\omega}, \qquad (4)$$

where we note that the vector $\boldsymbol{\omega}$ has zero mean and that, with a slight abuse of notation, $D(\boldsymbol{x})$ denotes the $|\mathcal{E}|$-dimensional diagonal matrix of the sine of the phase differences uniquely defined by the $(n-1)$-dimensional vector $\boldsymbol{x}_{\mathrm{desired}}$.

We begin by studying the problem of attaining a desired functional pattern using only nonnegative weights. With the above notation, for a desired functional pattern corresponding to the phase differences $\boldsymbol{x}$, this problem reads as

$$\text{find} \qquad \boldsymbol{\delta} \qquad (5)$$

$$\text{subject to} \qquad BD(\boldsymbol{x})\boldsymbol{\delta} = \boldsymbol{\omega}, \qquad (5a)$$

$$\text{and} \qquad \boldsymbol{\delta} \geq 0. \qquad (5b)$$

It should be noticed that the feasibility of the optimization problem (5) depends on the sign of the entries of the diagonal matrix $D(\boldsymbol{x})$, but is independent of their magnitude. To see this, notice that

$$D(\boldsymbol{x}) = \mathrm{sign}(D(\boldsymbol{x})) |D(\boldsymbol{x})|,$$

where the $\mathrm{sign}(\cdot)$ and absolute value $|\cdot|$ operators are applied element-wise. Then, Problem (5) is feasible if and only if there exists a nonnegative solution to

$$\underbrace{B\,\mathrm{sign}(D(\boldsymbol{x}))}_{\bar{B}} \underbrace{|D(\boldsymbol{x})|\boldsymbol{\delta}}_{\bar{\boldsymbol{\delta}}} = \boldsymbol{\omega}.$$

The feasibility of the latter equation, in turn, depends on the projections of the natural frequencies $\boldsymbol{\omega}$ on the columns of $\bar{B}$: a nonnegative solution exists if $\boldsymbol{\omega}$ belongs to the cone generated by the columns of $\bar{B}$. This also implies that, if a network admits a desired functional pattern $\boldsymbol{x}$ then, by tuning its weights, the same network can generate any other functional pattern $\boldsymbol{x}_{\mathrm{new}}$ such that $\mathrm{sign}(D(\boldsymbol{x}_{\mathrm{new}})) = \mathrm{sign}(D(\boldsymbol{x}))$. Thus, by properly tuning its weights, a network can generally generate a continuum of functional patterns determined uniquely by the signs of its incidence matrix and the oscillators natural frequencies. This property is illustrated in Fig. 2 for the case of a line network.

A sufficient condition for the feasibility of Problem (5) is as follows:

*There exists $\boldsymbol{\delta} \geq 0$ such that $BD(\boldsymbol{x})\boldsymbol{\delta} = \boldsymbol{\omega}$ if there exists a set $\mathcal{S}$ satisfying:*

(i.a) $D_{ii}(\boldsymbol{x})D_{jj}(\boldsymbol{x})B_{:,i}^\mathsf{T} B_{:,j} \leq 0$ *for all* $i, j \in \mathcal{S}$ *with* $i \neq j$ *and* $D_{ii}, D_{jj} \neq 0$;

(i.b) $\boldsymbol{\omega}^\mathsf{T} B_{:,i} D_{ii}(\boldsymbol{x}) > 0$ *for all* $i \in \mathcal{S}$;

(i.c) $\boldsymbol{\omega} \in \mathrm{Im}(B_{:,\mathcal{S}})$.

Equivalently, the above conditions ensure that $\boldsymbol{\omega}$ is contained within the cone generated by the columns of $\bar{B}_{:,\mathcal{S}}$ (see Fig. 3a for a self-contained example). To see this, rewrite the pattern assignment problem $BD(\boldsymbol{x})\boldsymbol{\delta} = \boldsymbol{\omega}$ as

$$BD(\boldsymbol{x})\boldsymbol{\delta} = B_{:,\mathcal{S}}D_{\mathcal{S},\mathcal{S}}(\boldsymbol{x})\boldsymbol{\delta}_{\mathcal{S}} + B_{:,\tilde{\mathcal{S}}}D_{\tilde{\mathcal{S}},\tilde{\mathcal{S}}}(\boldsymbol{x})\boldsymbol{\delta}_{\tilde{\mathcal{S}}} = \boldsymbol{\omega}, \qquad (6)$$

where the subscripts $\mathcal{S}$ and $\tilde{\mathcal{S}}$ denote the entries corresponding to the set $\mathcal{S}$ and the remaining ones, respectively. If the vectors $B_{:,i}$, $i \in \mathcal{S}$, are linearly independent, condition (i.a) implies that $D_{\mathcal{S},\mathcal{S}}B_{:,\mathcal{S}}^\mathsf{T} B_{:,\mathcal{S}}D_{\mathcal{S},\mathcal{S}}$ is an M-matrix; that is, a matrix which has nonpositive off-diagonal elements and positive principal minors[34]. Otherwise, the argument holds verbatim by replacing $\mathcal{S}$ with any subset $\mathcal{S}_{\mathrm{ind}} \subset \mathcal{S}$ such that the vectors $B_{:,i}$, $i \in \mathcal{S}_{\mathrm{ind}}$, are linearly independent. Condition (i.c) guarantees the existence of a solution to $B_{:,\mathcal{S}}D_{\mathcal{S},\mathcal{S}}\boldsymbol{\delta}_{\mathcal{S}} = \boldsymbol{\omega}$. A particular solution to the latter

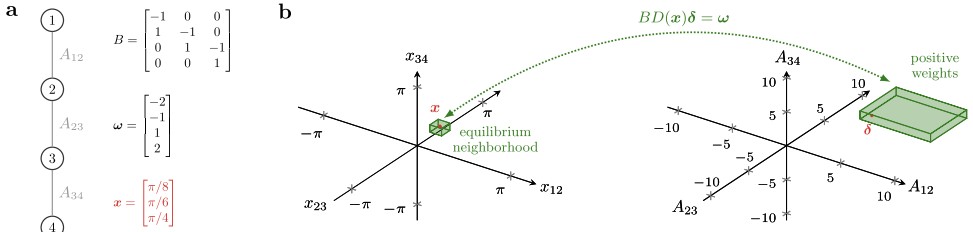

**Fig. 2 Mapping between desired phase differences and interconnection weights. a** A line network of $n = 4$ nodes and its parameters. The desired phase differences are shown in red. **b** Left panel: space of the phase differences; right panel: space of the interconnection weights. The pattern $\boldsymbol{x}$ is illustrated in red in the left panel, and the network weights that achieve such a pattern are represented in red in the right panel. For fixed natural frequencies $\boldsymbol{\omega}$, the green parallelepiped on the left represents the continuum of functional patterns within 0.2 radians from $\boldsymbol{x}$ which can be generated by the positive interconnection weights in the green parallelepiped on the right.

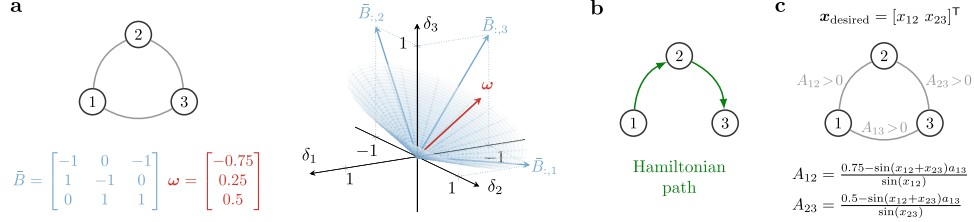

**Fig. 3 Algebraic and graph-theoretic conditions for the existence of positive weights that attain a desired functional pattern. a** The left side illustrates a simple network of 3 oscillators with adjacency matrix $\bar{B}$ and vector of natural frequencies $\boldsymbol{\omega}$. The right side illustrates the cone generated by the columns of $\bar{B}$. In this example, $\mathcal{S} = \{1, 2\}$ satisfies the conditions for the existence of $\boldsymbol{\delta} \geq 0$ in Eq. (5), as $\boldsymbol{\omega}$ is contained within the cone generated by the columns $\bar{B}_{:,\mathcal{S}}$. **b** The (directed) Hamiltonian path described by the columns of $\bar{B}_{:,\mathcal{H}}$, with $\mathcal{H} = \{1, 2\}$, in the network of panel (**a**). **c** The existence of such an Hamiltonian path, together with a positive projection of $\boldsymbol{\omega}$ onto $\bar{B}_{:,\mathcal{H}}$, also ensure that there exists a strictly positive $\boldsymbol{\delta} > 0$ solution to $BD(\boldsymbol{x})\boldsymbol{\delta} = \boldsymbol{\omega}$. In particular, for any choice of $x_{12}, x_{23} \in (0, \pi)$, Eq. (4) reveals that if $0 < A_{13} < 0.5/\sin(x_{12} + x_{23})$, then there exist strictly positive weights $A_{12} > 0$ and $A_{23} > 0$ such that $\boldsymbol{\delta} > 0$.

equation is

$$\boldsymbol{\delta}_{\mathcal{S}} = (B_{:,\mathcal{S}}D_{\mathcal{S},\mathcal{S}}(\boldsymbol{x}))^{\dagger}\boldsymbol{\omega} = (D_{\mathcal{S},\mathcal{S}}(\boldsymbol{x})B_{:,\mathcal{S}}^{\mathsf{T}}B_{:,\mathcal{S}}D_{\mathcal{S},\mathcal{S}}(\boldsymbol{x}))^{-1}D_{\mathcal{S},\mathcal{S}}(\boldsymbol{x})B_{:,\mathcal{S}}^{\mathsf{T}}\boldsymbol{\omega} > 0$$

where $(\cdot)^{\dagger}$ denotes the Moore–Penrose pseudo-inverse of a matrix. The positivity of $\boldsymbol{\delta}_{\mathcal{S}}$ follows from condition (i.b) and the fact that the inverse of an $M$-matrix is element-wise nonnegative[34]. We conclude that a solution to Eq. (6) is given by $\boldsymbol{\delta}_{\mathcal{S}} = (B_{:,\mathcal{S}}D_{\mathcal{S},\mathcal{S}}(\boldsymbol{x}))^{\dagger}\boldsymbol{\omega} > 0$ and $\boldsymbol{\delta}_{\bar{\mathcal{S}}} = 0$.

To avoid disconnecting edges or to maintain a fixed network topology, a functional pattern should be realized in Problem (5) using a strictly positive weight vector (that is, $\boldsymbol{\delta} > 0$ rather than $\boldsymbol{\delta} \geq 0$). While, in general, this is a considerably harder problem, a sufficient condition for the existence of a strictly positive solution $\boldsymbol{\delta} > 0$ is that the network with incidence matrix $\bar{B}$ contains an Hamiltonian path, that is, a directed path that visits all the oscillators exactly once (Fig. 3b shows a network containing an Hamiltonian path). Namely,

*There exists a strictly positive solution $\boldsymbol{\delta} > 0$ to $BD(\boldsymbol{x})\boldsymbol{\delta} = \boldsymbol{\omega}$ if*

(ii.a) *the network with incidence matrix $\bar{B}$ contains a directed Hamiltonian path $\mathcal{H}$;*

(ii.b) *$\boldsymbol{\omega}^{\mathsf{T}}\bar{B}_{:,i}D_{ii}(\boldsymbol{x}) > 0$ for all $i \in \mathcal{H}$;*

The incidence matrix $\bar{B}_{:,\mathcal{H}}$ of a directed Hamiltonian path $\mathcal{H}$ has two key properties. First, it comprises $n - 1$ linearly independent columns, since the path covers all the vertices and contains no cycles. This guarantees that $\boldsymbol{\omega} \in \mathrm{Im}(B_{:,\mathcal{H}})$. Second, the columns of the incidence matrix $\bar{B}_{:,\mathcal{H}}$ satisfy $\bar{B}_{:,i}^{\mathsf{T}}\bar{B}_{:,i} = 2$ and $\bar{B}_{:,i}^{\mathsf{T}}\bar{B}_{:,j} \in \{0, -1\}$ for all $i, j \in \mathcal{H}$, $i \neq j$. Then, letting the set $\mathcal{S}$ in the result above identify the columns of the Hamiltonian path, conditions (ii.a) and (ii.b) imply (i.a)–(i.c), thus ensuring the existence of a nonnegative set of weights $\boldsymbol{\delta}$ that solves the pattern assignment problem $BD(\boldsymbol{x})\boldsymbol{\delta} = \boldsymbol{\omega}$. Furthermore, by rewriting the pattern assignment problem as in Eq. (6), the following vector of

interconnection weights solves such an equation (see the "Methods" section):

$$\boldsymbol{\delta}_{\mathcal{H}} = (B_{:,\mathcal{H}}D_{\mathcal{H},\mathcal{H}}(\boldsymbol{x}))^{\dagger}(\boldsymbol{\omega} - B_{:,\tilde{\mathcal{H}}}D_{\tilde{\mathcal{H}},\tilde{\mathcal{H}}}(\boldsymbol{x})\boldsymbol{\delta}_{\tilde{\mathcal{H}}}).$$

Because $\bar{B}_{:,\mathcal{H}} = B_{:,\mathcal{H}}D_{\mathcal{H},\mathcal{H}}$ defines an Hamiltonian path and because of (ii.b), the vector $(B_{:,\mathcal{H}}D_{\mathcal{H},\mathcal{H}}(\boldsymbol{x}))^{\dagger}\boldsymbol{\omega}$ contains only strictly positive entries. Thus, for any sufficiently small and positive vector $\boldsymbol{\delta}_{\tilde{\mathcal{H}}}$, the weights $\boldsymbol{\delta}_{\mathcal{H}}$ are also strictly positive, ultimately proving the existence of a strictly positive solution to the pattern assignment problem (see the "Methods" section). Figure 3c illustrates a self-contained example.

Taken together, the results presented in this section reveal that the interplay between the network structure and the oscillators' natural frequencies dictates whether a desired functional pattern is achievable under the constraint of nonnegative (or even strictly positive) interconnections. First, dense positive networks with a large number of edges are more likely to generate a desired functional pattern, since their incidence matrix features a larger number of candidate vectors to satisfy conditions (i.a)–(i.c). Second, densely connected networks are also more likely to contain an Hamiltonian path, thus promoting also strictly positive network designs. Third, after an appropriate relabeling of the oscillators such that any interconnection from $i$ to $j$ in the Hamiltonian path satisfies $i < j$, condition (ii.b) is equivalent to requiring that $\omega_i < \omega_j$. That is, the feasibility of a functional pattern is guaranteed when the natural frequencies increase monotonically with the ordering identified by the Hamiltonian path. This also implies, for instance, that sparsely connected positive networks, and not only dense ones, can attain a large variety of functional patterns. An example is a connected line network with increasing natural frequencies, which can generate, among others, any functional pattern defined by phase differences that are smaller than $\frac{\pi}{2}$ (trivially, when the phase differences are

smaller than $\frac{\pi}{2}$ and the natural frequencies are increasing, a line network contains an Hamiltonian path and the vector of natural frequencies has positive projections onto the columns of the incidence matrix). Figure 2a contains an example of such a network.

**Compatibility of multiple functional patterns.** A single choice of the interconnections weights can allow for multiple desired functional patterns, as long as they are compatible with the network dynamics in Eq. (1). In this section, we provide a characterization of compatible functional patterns in a given network, and derive conditions for the existence of a set of interconnection weights that achieve multiple desired functional patterns. Being able to concurrently assign multiple functional patterns is crucial, for instance, to the investigation and design of memory systems[35], where different patterns of activity correspond to distinct memories. Furthermore, our results complement previous work on the search for equilibria in oscillator networks[36].

To find a set of functional patterns that exist concurrently in a given network with fixed interconnection weights $\boldsymbol{\delta}$, we exploit the algebraic core of Eq. (4) and show that the kernel of the incidence matrix $B$ uniquely determines the equilibria of the network. In fact, for a given network (i.e., $\boldsymbol{\delta}$ with nonzero components) all compatible equilibria $\boldsymbol{x}^{(i)}$, $i \in \{1, \ldots, \ell\}$, must satisfy

$$D(\boldsymbol{x}^{(i)})\boldsymbol{\delta} = B^{\dagger}\boldsymbol{\omega} + \ker(B). \tag{7}$$

From Eq. (7), we can see that the sine vector of all compatible equilibria must belong to a specific affine subspace of $\mathbb{R}^{|\mathcal{E}|}$:

$$\sin(\boldsymbol{x}^{(i)}) \in \operatorname{diag}(\boldsymbol{\delta})^{-1}\left(B^{\dagger}\boldsymbol{\omega} + \ker(B)\right). \tag{8}$$

Rewriting Eq. (4) in the above form connects the existence of distinct functional patterns with $\ker(B)$, the latter featuring a number of well-known properties. For instance, it holds that $\dim(\ker(B)) = |\mathcal{E}| - n + c$, where $c$ is the number of connected components in a network, and that $\ker(B)$ coincides with the subspace spanned by the signed path vectors of all undirected cycles in the network[37]. Notice also that, after a suitable reordering of the phase differences in $\boldsymbol{x}$, we can write $\sin(\boldsymbol{x}) = [\sin(\boldsymbol{x}_{\text{desired}}^{\mathsf{T}})\, \sin(\boldsymbol{x}_{\text{dep}}^{\mathsf{T}})]^{\mathsf{T}}$, where $\boldsymbol{x}_{\text{dep}}$ denotes the phase differences dependent on $n-1$ desired phase differences $\boldsymbol{x}_{\text{desired}}$. Thus, all the $\boldsymbol{x}_{\text{desired}}$ for which $\sin(\boldsymbol{x}_{\text{dep}})$ intersects the affine space described by $\operatorname{diag}(\boldsymbol{\delta})^{-1}(B^{\dagger}\boldsymbol{\omega} + \ker(B))$ identify compatible functional patterns.

To showcase how the intimate relationship between the network structure and the kernel of its incidence matrix enables the characterization of which (and how many) compatible patterns coexist, we consider three essential network topologies: trees, cycles, and complete graphs. For the sake of simplicity, we let $\boldsymbol{\delta} = \boldsymbol{1}$ and $\boldsymbol{\omega} = 0$, so that Eq. (8) holds whenever $\sin(\boldsymbol{x}^{(i)}) \in \ker(B)$. In networks with tree topologies it holds $\ker(B) = 0$, and $\sin(\boldsymbol{x}^{(i)}) = 0$ is satisfied by $2^{n-1}$ patterns of the form $x_{jk}^{(i)} = 0, \pi$, for all $(j,k) \in \mathcal{E}$. Consider now cycle networks, where $\ker(B) = \operatorname{span}\boldsymbol{1}$. For any cycle of $n \geq 3$ oscillators, two families of patterns are straightforward to derive. First, there are $2^{n-1}$ patterns of the form $x_{k,k+1}^{(i)} = 0, \pi$, with $k = 1, \ldots, n-1$, and $x_{n1}^{(i)} = -\sum_{k=1}^{n-1} x_{k,k+1}^{(i)}$. Second, there are $n-1$ splay states[17], where the oscillators' phases evenly span the unitary circle, with $x_{jk}^{(i)} = \frac{2\pi m}{n}$, $m = 1, \ldots, n-1$, $(j,k) \in \mathcal{E}$. Figure 4 illustrates the compatible functional patterns satisfying Eq. (8) in a positive network of three fully synchronizing oscillators. In general, however, cycle networks of identical oscillators admit infinite coexisting patterns. For instance, Fig. 5 shows how we can parameterize infinite equilibria with a scalar $\gamma \in \mathbb{S}^1$ in a cycle of $n = 4$ oscillators. Finally, as complete graphs are equivalent to a composition of cycles, they also admit infinite compatible patterns that can be parameterized akin to what occurs in a simple cycle (see Supplementary Fig. 2).

We now turn our attention to finding the interconnection weights that simultaneously enable a collection of $\ell \geq 1$ desired functional patterns $\{\boldsymbol{x}^{(i)}\}_{i=1}^{\ell}$. We first notice that Eq. (7) reveals that to achieve a desired functional pattern $\boldsymbol{x}^{(i)}$ with components not equal to $k\pi$, $k \in \mathbb{Z}$, the network weights $\boldsymbol{\delta}$ must belong to an $|\mathcal{E}|$-dimensional affine subspace of $\mathbb{R}^{|\mathcal{E}|}$:

$$\boldsymbol{\delta} \in D(\boldsymbol{x}^{(i)})^{-1}(B^{\dagger}\boldsymbol{\omega} + \ker(B)), \quad \forall i = 1, \ldots, \ell. \tag{9}$$

Let $\Gamma_i = D(\boldsymbol{x}^{(i)})^{-1}(B^{\dagger}\boldsymbol{\omega} + \ker(B))$. Then, to concurrently realize a collection of patterns $\{\boldsymbol{x}^{(i)}\}_{i=1}^{\ell}$, a solution to Eq. (9) exists if and only if $\bigcap_{i=1}^{\ell} \Gamma_i \neq \emptyset$. It is worth noting that, whenever the latter intersection coincides with a singleton, then there exists a single choice of network weights that realizes $\{\boldsymbol{x}^{(i)}\}_{i=1}^{\ell}$. However, if $\bigcap_{i=1}^{\ell} \Gamma_i$ corresponds to a subspace, then infinite networks can realize the desired collection of functional patterns. We conclude by emphasizing that a positive $\boldsymbol{\delta}$ that achieves the desired patterns exists if and only if $(\bigcap_{i=1}^{\ell} \Gamma_i) \cap \mathbb{R}_{\geq 0}^{|\mathcal{E}|} \neq \emptyset$. That is, if the network weights belong to the nonempty intersection of the $\ell$ affine subspaces with the positive orthant.

**Stability of functional patterns.** A functional pattern is stable when small deviations of the oscillators phases from the desired configuration lead to vanishing functional perturbations. Stability is a desired property since it guarantees that the desired functional pattern is robust against perturbations to the oscillators dynamics. To study the stability of a functional pattern, we analyze the Jacobian of the Kuramoto dynamics at the desired functional configuration, which reads as[17]

$$J = \frac{\partial}{\partial \boldsymbol{\theta}}\dot{\boldsymbol{\theta}} = -\underbrace{B\operatorname{diag}\left(\{A_{ij}\cos(x_{ij})\}_{(i,j)\in\mathcal{E}}\right)B^{\mathsf{T}}}_{\mathcal{L}(\boldsymbol{x})}, \tag{10}$$

where $\mathcal{L}(\boldsymbol{x})$ denotes the Laplacian matrix of the network with weights scaled by the cosines of the phase differences (the weight between nodes $i$ and $j$ is $A_{ij}\cos(\theta_i - \theta_j)$). The functional pattern $\boldsymbol{x}$ is stable when the eigenvalues of the above Jacobian matrix have negative real parts. For instance, if all phase differences are strictly smaller than $\frac{\pi}{2}$ (that is, the infinity-norm of $\boldsymbol{x}$ satisfies $\|\boldsymbol{x}\|_{\infty} < \frac{\pi}{2}$), then the Jacobian in Eq. (10) is known to be stable[17]. In the case that both cooperative and competitive interactions are allowed, we can ensure stability of a desired pattern by specifying the network weights in $\boldsymbol{\delta}$ such that $A_{ij} > 0$ if $|x_{ij}| < \frac{\pi}{2}$ and $A_{ij} < 0$ otherwise, so that the matrix $\mathcal{L}$ becomes the Laplacian of a positive network (see Methods). Furthermore, we observe that in the particular case where some differences $|x_{ij}| = \frac{\pi}{2}$, the network may become disconnected since $\cos(x_{ij}) = 0$. Because the Laplacian of a disconnected network has multiple eigenvalues at zero, marginal stability may occur, and phase trajectories may not converge to the desired pattern.

When some phase differences are larger than $\frac{\pi}{2}$ and the network allows only for nonnegative weights, then stability of a functional pattern is more difficult to assess because the Jacobian matrix becomes a *signed* Laplacian[38]. The off-diagonal entries of a signed Laplacian satisfy $\mathcal{L}_{ij} > 0$ whenever $|x_{ij}| > \frac{\pi}{2}$, thus possibly changing the sign of its diagonal entries $\mathcal{L}_{ii} = \sum_j A_{ij}\cos(x_{ij})$ and violating the conditions for the use of classic results from algebraic graph theory for the stability of Laplacian matrices. To derive a

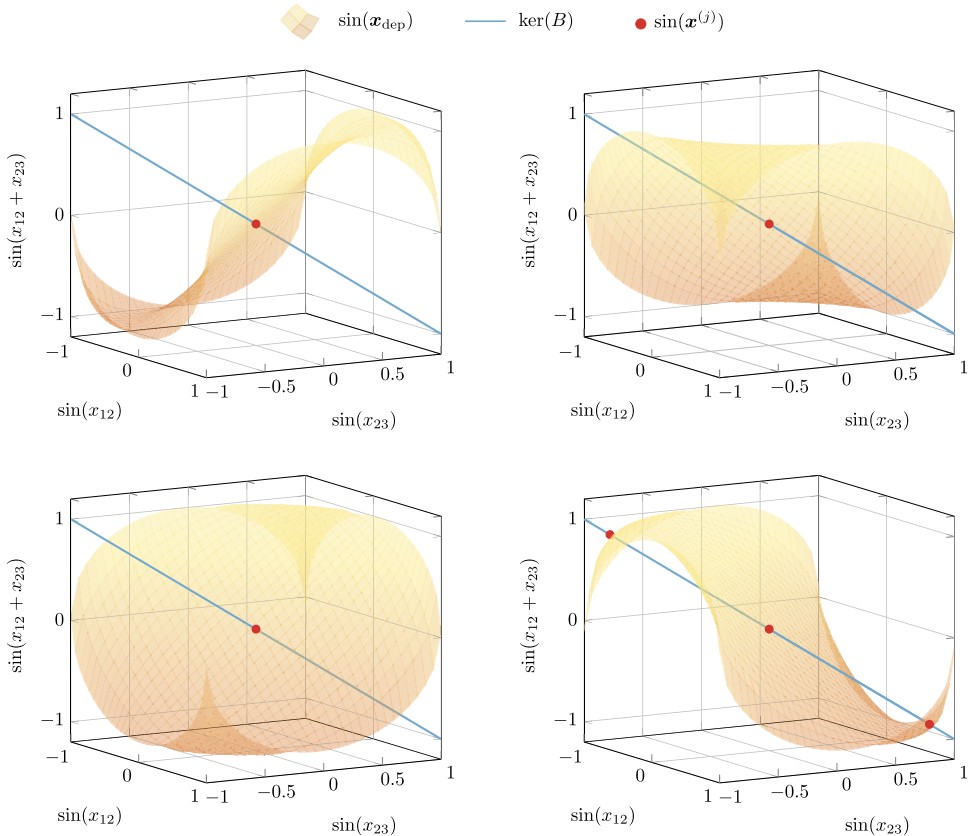

**Fig. 4 The intersection of an affine space with $\sin(\boldsymbol{x}_{\text{dep}})$ determines the compatible functional patterns of 3 identical oscillators.** Consider a fully connected network of $n = 3$ identical oscillators with zero natural frequency and $\boldsymbol{\delta} = \mathbf{1}$. It is well known that $\boldsymbol{x}^{(1)} = [0\ 0]^{\mathsf{T}}$, $\boldsymbol{x}^{(2)} = [\pi\ 0]^{\mathsf{T}}$, $\boldsymbol{x}^{(3)} = [0\ \pi]^{\mathsf{T}}$, $\boldsymbol{x}^{(4)} = [\pi\ \pi]^{\mathsf{T}}$ are phase difference equilibria. Furthermore, because $\sin(\theta) = \sin(\pi - \theta)$, this figure illustrates $\sin(x_{13})$ as a function of $x_{12}$ and $x_{23}$ in four different panels: $\sin(x_{12} + x_{23})$ (top left), $\sin(\pi - x_{12} + x_{23})$ (top right), $\sin(x_{12} + \pi - x_{23})$ (bottom left), and $\sin(-x_{12} - x_{23})$ (bottom right). The fourth panel reveals that the two functional patterns compatible with $\boldsymbol{x}^{(j)}$, $j \in \{1, ..., 4\}$, correspond to $\boldsymbol{x}^{(5)} = [2\pi/3\ 2\pi/3]^{\mathsf{T}}$ and $\boldsymbol{x}^{(6)} = [-2\pi/3\ -2\pi/3]^{\mathsf{T}}$ (in red).

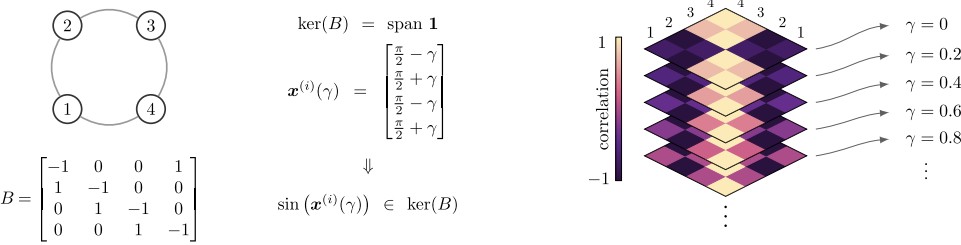

**Fig. 5 A homogeneous cycle network admits infinite compatible functional patterns.** Since $\ker(B) = \operatorname{span} \mathbf{1}$, the cycle network admits infinite compatible equilibria, which can be parameterized by $\gamma \in \mathbb{S}^1$ as $\boldsymbol{x}^{(i)}(\gamma) = [\frac{\pi}{2} - \gamma, \frac{\pi}{2} + \gamma, \frac{\pi}{2} - \gamma, \frac{\pi}{2} + \gamma]^{\mathsf{T}}$. Any arbitrarily small variation of $\gamma$ yields $\sin(\boldsymbol{x}^{(i)}(\gamma)) \in \ker(B)$. The right panel illustrates the patterns associated with $\boldsymbol{x}^{(i)}(\gamma)$, $i = 1, ..., 5$ for increments of $\gamma$ of 0.2 radians.

condition for the instability of the Jacobian in Eq. (10), we exploit the notion of *structural balance*. We say that the cosine-scaled network with Laplacian matrix $\mathcal{L}$ is structurally balanced if and only if its oscillators can be partitioned into two sets, $\mathcal{O}_1$ and $\mathcal{O}_2$, such that all $(i, j) \in \mathcal{E}$ with $A_{ij} \cos(x_{ij}) < 0$ connect oscillators in $\mathcal{O}_1$ to oscillators in $\mathcal{O}_2$, and all $(i, j) \in \mathcal{E}$ with $A_{ij} \cos(x_{ij}) > 0$ connect oscillators within $\mathcal{O}_i$, $i \in \{1, 2\}$. If a network is structurally balanced, then its Laplacian has mixed eigenvalues[38]. Therefore, we conclude the following:

*If the functional pattern $\boldsymbol{x}$ yields a structurally balanced cosine-scaled network, then $\boldsymbol{x}$ is unstable.*

The above condition allows us to immediately assess the instability of functional patterns for the special cases of line and cycle networks. In fact, for a line network with positive weights, $\boldsymbol{x}$ is unstable whenever $|x_{ij}| > \frac{\pi}{2}$ for any $i, j$. Instead, for a cycle network with positive weights, the pattern $\boldsymbol{x}$ can be stable only if it contains at most one phase difference $\frac{\pi}{2} < |x_{ij}| < \gamma$, where $\gamma \approx 1.789776$ solves $\gamma - \tan(\gamma) = 2\pi$ (see Supplementary Information). In the next section, we propose a heuristic procedure to correct the interconnection weights in positive networks to promote stability of a functional pattern.

**Optimal interventions for desired functional patterns.** Armed with conditions to guarantee the existence of positive interconnections that enable a desired functional pattern, we now show that the problem of adjusting the network weights to generate a desired functional pattern can be cast as a convex optimization problem. Formally, for a desired functional pattern $\boldsymbol{x}$

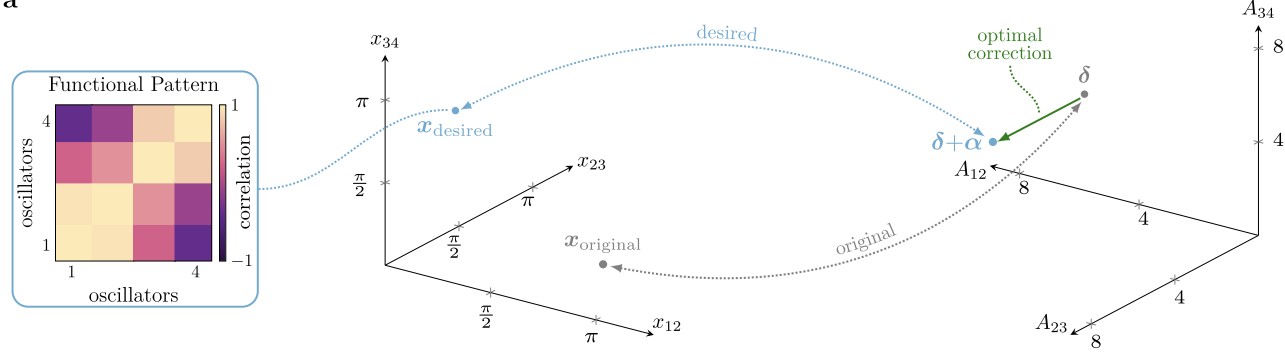

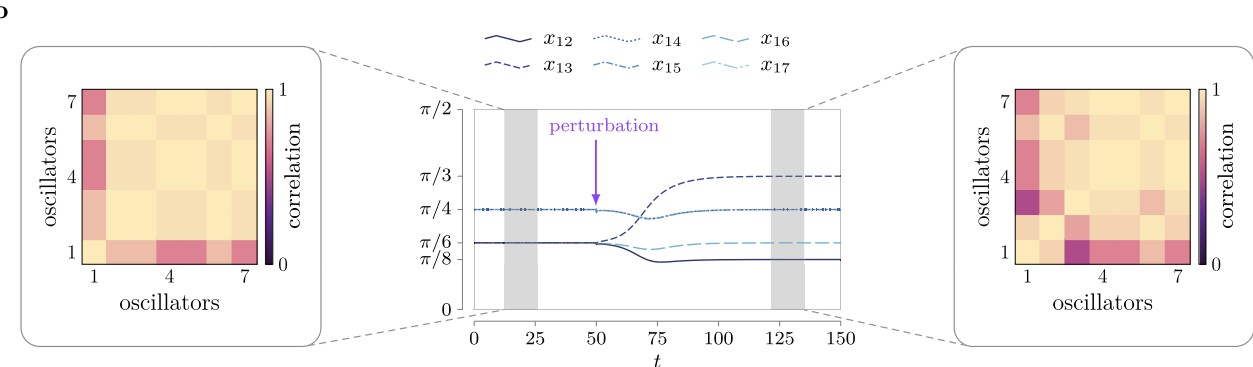

**Fig. 6 Optimal interventions for desired functional patterns. a** For the line network in Fig. 2a, we solve Problem (11) to assign the desired pattern $\boldsymbol{x}_{\text{desired}} = [\frac{4\pi}{5}\,\frac{\pi}{3}\,\frac{\pi}{10}]^{\mathsf{T}}$. The starting pattern $\boldsymbol{x}_{\text{original}} = [\frac{\pi}{10}\,\frac{\pi}{3}\,\frac{4\pi}{5}]^{\mathsf{T}}$ is associated with interconnection weights $\boldsymbol{\delta} = [3.4026\ 3.4641\ 6.4721]^{\mathsf{T}}$. Applying the optimal correction $\boldsymbol{\alpha}^*$ yields positive interconnection weights $\boldsymbol{\delta} + \boldsymbol{\alpha}^* = [6.4721\ 3.4026\ 3.4641]^{\mathsf{T}}$ that achieve the desired functional patterns $\boldsymbol{x}_{\text{desired}}$. **b** Joint allocation of two compatible equilibria for the phase difference dynamics. By taking $\theta_1$ as a reference, we choose two points for the phase differences $x_{1i} = \theta_i - \theta_1$, $i \in \{2, ..., 7\}$, to be imposed as equilibria in a network of $n = 7$ oscillators: $\boldsymbol{x}_{\text{desired}}^{(1)} = \left[\frac{\pi}{6}\,\frac{\pi}{6}\,\frac{\pi}{4}\,\frac{\pi}{4}\,\frac{\pi}{6}\,\frac{\pi}{4}\right]^{\mathsf{T}}$ and $\boldsymbol{x}_{\text{desired}}^{(2)} = \left[\frac{\pi}{8}\,\frac{\pi}{3}\,\frac{\pi}{4}\,\frac{\pi}{4}\,\frac{\pi}{6}\,\frac{\pi}{4}\right]^{\mathsf{T}}$. In this example, we find a set of interconnection weights $(\boldsymbol{\delta} + \boldsymbol{\alpha}^*)$ that solves the minimization problem (11) with constraint (12). The trajectories start at the (unstable) equilibrium point $\boldsymbol{x}_{\text{desired}}^{(1)}$ at time $t = 0$, and converge to the point $\boldsymbol{x}_{\text{desired}}^{(2)}$ after a small perturbation $\boldsymbol{p} \in \mathbb{T}^7$, with $\pi \in [0\ 0.05]$, is applied to the phase difference trajectories at time $t = 50$.

and network weights $\boldsymbol{\delta}$, we seek to solve

$$\min_{\boldsymbol{\alpha}} \quad \|\boldsymbol{\alpha}\|_2 \tag{11}$$

$$\text{subject to} \quad BD(\boldsymbol{x})(\boldsymbol{\delta} + \boldsymbol{\alpha}) = \boldsymbol{\omega}, \tag{11a}$$

$$\text{and} \quad (\boldsymbol{\delta} + \boldsymbol{\alpha}) \geq 0, \tag{11b}$$

where $\boldsymbol{\alpha} \in \mathbb{R}^{|\mathcal{E}|}$ are the controllable modifications of the network weights, and $\|\cdot\|_2$ denotes the $\ell^2$-norm. Figure 6a illustrates the control of a functional pattern in a line network of $n = 4$ oscillators.

The minimization problem (11) is convex, thus efficiently solvable even for large networks, and may admit multiple minimizers, thus showing that different networks may exhibit the same functional pattern. Moreover, in light of our results above, Problem (11) can be easily adapted to assign a collection of desired patterns $\{\boldsymbol{x}^{(i)}\}_{i=1}^{\ell}$. To do so, we simply replace the constraint (11a) with

$$BD(\boldsymbol{x}^{(i)})(\boldsymbol{\delta} + \boldsymbol{\alpha}) = \boldsymbol{\omega}, \quad \forall i = 1, \ldots, \ell. \tag{12}$$

Figure 6b illustrates an example where we jointly allocate two functional patterns for a complete graph with $n = 7$ oscillators (see Supplementary Information for more details on this example).

Note that the minimization problem (11) does not guarantee that the functional pattern $\boldsymbol{x}$ is stable for the network with weights

$\boldsymbol{\delta} + \boldsymbol{\alpha}^*$. To promote stability of the pattern $\boldsymbol{x}$, we use a heuristic procedure based on the classic Gerschgorin's theorem[39]. Recall that the stability of $\boldsymbol{x}$ is guaranteed when the associated Jacobian matrix has a Laplacian structure, with negative diagonal entries and nonnegative off-diagonal entries. Further, instability of $\boldsymbol{x}$ depends primarily on the negative off-diagonal entries $A_{ij}\cos(x_{ij})$ of the Jacobian (these entries are negative because the sign of the network weight $A_{ij}$ is different from the sign of the cosine of the desired phase difference $x_{ij}$). Therefore, reducing the magnitude of such entries $A_{ij}$ heuristically moves the eigenvalues of the Jacobian towards the stable half of the complex plane (this phenomenon can be captured using the Gerschgorin circles, as we show in Fig. 7 for a network with 7 nodes). To formalize this procedure, let $\boldsymbol{\delta}_{\mathcal{N}}$ and $\boldsymbol{\alpha}_{\mathcal{N}}$ denote the entries of the weights $\boldsymbol{\delta}$ and tuning vector $\boldsymbol{\alpha}$, respectively, that are associated to negative interconnections $A_{ij}\cos(x_{ij}) < 0$ in the cosine-scaled network. Then, the optimization problem that enacts the proposed strategy becomes:

$$\min_{\boldsymbol{\alpha}} \quad \|\boldsymbol{\delta}_{\mathcal{N}} + \boldsymbol{\alpha}_{\mathcal{N}}\|_2$$
$$\text{subject to} \quad BD(\boldsymbol{x})(\boldsymbol{\delta} + \boldsymbol{\alpha}) = \boldsymbol{\omega}, \tag{13}$$
$$\text{and} \quad (\boldsymbol{\delta} + \boldsymbol{\alpha}) \geq 0.$$

Carefully reducing the weights $\boldsymbol{\delta}_{\mathcal{N}} + \boldsymbol{\alpha}_{\mathcal{N}}$ promotes stability of the target pattern. Figure 7 illustrates the shift of the Jacobian's

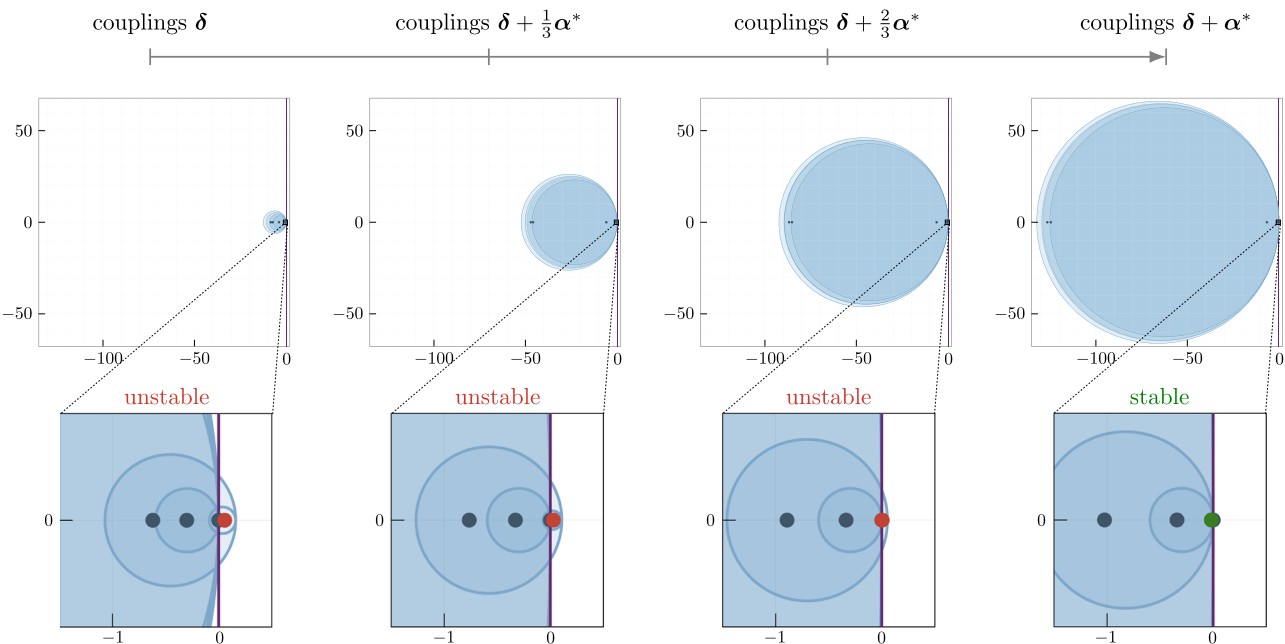

**Fig. 7 Mechanism underlying the heuristic procedure to promote stability of functional patterns containing negative correlations.** For the 7-oscillator network in Supplementary Text 1.5, we apply the procedure in equation (13) to achieve stability of the pattern $x_{\text{desired}} = \left[\frac{21\pi}{32} \frac{\pi}{6} \frac{\pi}{6} \frac{\pi}{8} \frac{\pi}{8} \frac{\pi}{3}\right]^{\mathsf{T}}$, where $x_{12} = \theta_2 - \theta_1 > \frac{\pi}{2}$. The left plot illustrates the Gerschgorin disks (in blue) and the Jacobian's eigenvalues locations for the original network (as dark dots). The complex axis is highlighted in purple. It can be observed in the zoomed-in panel that one eigenvalue is unstable ($\lambda_2 = 0.0565$, in red). The optimal correction $\alpha^\star$ is gradually applied to the existing interconnections from the left-most panel to the right-most one at $\frac{1}{3}$ increments. The right zoomed-in panel shows that, as a result of our procedure, $n-1$ eigenvalues ultimately lie in the left-hand side of the complex plane ($\lambda_1 = 0$ due to rotational symmetry and $\lambda_2 = -0.0178$, in green).

eigenvalues while the optimal tuning vector $\boldsymbol{\alpha}^\star$ is gradually applied to a 7-oscillator network to achieve stability of a functional pattern containing negative correlations (the network parameters can be found in the Supplementary Information). Finally, we remark that the procedure in Eq. (13) can be further refined by introducing scaling constants to penalize $\left\|\boldsymbol{\delta}_{\mathcal{N}} + \boldsymbol{\alpha}_{\mathcal{N}}\right\|_2$ differently from the modification of other interconnection weights (see Supplementary Information for further details and an example).

The minimization problems (11) and (13) do not allow us to tune the oscillators' natural frequencies, and are constrained to networks with positive weights. When any parameter of the network is unconstrained and can be adjusted to enforce a desired functional pattern, the network optimization problem can be generalized as

$$\min_{\boldsymbol{\alpha}, \boldsymbol{\beta}} \quad \left\| \begin{bmatrix} \boldsymbol{\alpha}^{\mathsf{T}} & \boldsymbol{\beta}^{\mathsf{T}} \end{bmatrix} \right\|_2 \tag{14}$$

$$\text{subject to} \quad BD(\boldsymbol{x})(\boldsymbol{\delta} + \boldsymbol{\alpha}) = [\omega_1 \cdots \omega_n]^{\mathsf{T}} + \boldsymbol{\beta}, \tag{14a}$$

where $\boldsymbol{\beta}$ denotes the correction to the natural frequencies. Problem (14) always admits a solution because $\boldsymbol{\beta}$ can be chosen to satisfy the constraint (14a) for any choice of the network parameters $\boldsymbol{\delta} + \boldsymbol{\alpha}$. Further, the (unique) solution to the minimization problem (14) can also be computed in closed form:

$$\begin{bmatrix} \boldsymbol{\alpha}^\star \\ \boldsymbol{\beta}^\star \end{bmatrix} = \begin{bmatrix} BD(\boldsymbol{x}) & -I_n \end{bmatrix}^{\dagger} ([\omega_1 \cdots \omega_n]^{\mathsf{T}} - BD(\boldsymbol{x})\boldsymbol{\delta})$$

where $I_n$ denotes the $n \times n$ identity matrix.

We conclude this section by noting that the minimization problems (11)–(14) can be readily extended to include other vector norms besides the $\ell_2$-norm in the cost function (e.g., the $\ell_1$-norm to promote sparsity of the corrections), and to be

applicable to directed networks. The latter extension can be attained by replacing the constraints (11a) and (14a) with a suitable rewriting of the matrix form (4). We refer the interested reader to the Supplementary Information for a comprehensive treatment of this extension and an example.

## Applications to complex networks
In the remainder of this paper, we apply our methods to an empirically reconstructed brain network and to the IEEE 39 New England power distribution network. In the former case, we use the Kuramoto model to map structure to function and find that local metabolic changes underlie the emergence of functional patterns of recorded neural activity. In the latter case, we use our methods to restore the nominal network power flow after a fault.

**Local metabolic changes govern the emergence of distinct functional patterns in the brain.** The brain can be studied as a network system in which Kuramoto oscillators approximate the rhythmic activity of different brain regions[12,30,40,41]. Over short time frames, the brain is capable of exhibiting a rich repertoire of functional patterns while the network structure and the interconnection weights remain unaltered. Functional patterns of brain activity not only underlie multiple cognitive processes but can also be used as biomarkers for different psychiatric and neurological disorders[42].

To shed light on the structure–function relationship of the human brain, we utilize Kuramoto oscillators evolving on an empirically reconstructed brain network. We hypothesize that the intermittent emergence of diverse patterns stems from changes in the oscillators' natural frequencies—which can be thought of as endogenous changes in metabolic regional activity regulated by glial cells[43] or exogenous interventions to modify undesired synchronization patterns[30]. First, we show that phase-locked trajectories of the Kuramoto model in Eq. (1) can be accurately

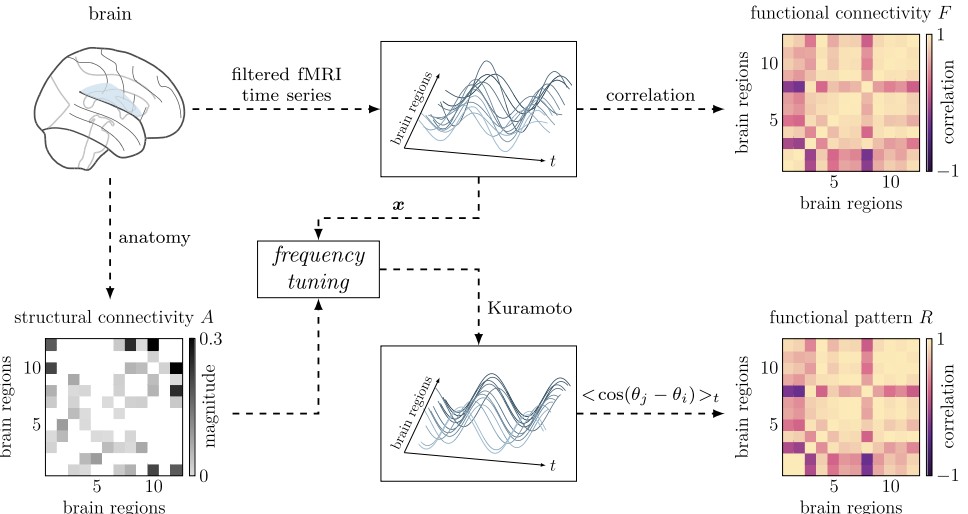

**Fig. 8 Replication of empirically recorded functional connectivity in the brain through tuning of the natural frequencies of Kuramoto oscillators.** The anatomical brain organization provides the network backbone over which the oscillators evolve. The filtered fMRI time series provide the relative phase differences between co-fluctuating brain regions, and thus define the desired phase differences $\boldsymbol{x}$, which is used to calculate the metabolic change encoded in the oscillators' natural frequencies. In this figure, we select the 40-s time window from $t_0 = 498$ s to $t_f = 538$ s for subject 18 in the second scanning session. We obtain $\|R-F\|_2 = 0.2879$. Additionally, we verify that the analysis of the Jacobian spectrum (see Eq. (10)) accurately predicts the stability of the phase-locked trajectories. Supplementary Fig. 7a illustrates the basin of attraction of $R$, which we numerically estimate to be half of the torus.

extracted from noisy measurements of neural activity and are a relatively accurate approximation of empirical data.

We employ structural (i.e., interconnections between brain regions) and functional (i.e., time series of recorded neural activity) data from ref. [40]. Structural connectivity consists of a sparse weighted matrix $A$ whose entries represent the strength of the physical interconnection between two brain regions. Functional data comprise time series of neural activity recorded through functional magnetic resonance imaging (fMRI) of healthy subjects at rest. Because the phases of the measured activity have been shown to carry most of the information contained in the slow oscillations recorded through fMRI time series, we follow the steps in ref. [40] to obtain such phases from the data by filtering the time series in the $[0.04, 0.07]$ Hz frequency range (Supplementary Information). Next, since frequency synchronization is thought to be a crucial enabler of information exchange between different brain regions and homeostasis of brain dynamics[44,45], we focus on functional patterns that arise from phase-locked trajectories, as compatible with our analysis. For simplicity, we restrict our study to the cingulo-opercular cognitive system, which, at the spatial scale of our data, comprises $n = 12$ interacting brain regions[46].

We identify time windows in the filtered fMRI time series where the signals are phase-locked, and compute two matrices for each time window: a matrix $F \in \mathbb{R}^{12 \times 12}$ of Pearson correlation coefficients (also known as functional connectivity), and a functional pattern $R \in \mathbb{R}^{12 \times 12}$ (as in equation (2)) from the phases extracted by solving the *nonconvex phase synchronization* problem[47]. Strikingly, we find that $\|F-R\|_2 \ll 1$ consistently (see Supplementary Information and Supplementary Fig. 7b), thus demonstrating that our definition of the functional pattern is a good replacement for the classical Pearson correlation arrangements in networks with oscillating states.

Building upon the above finding, we test whether the oscillators' natural frequencies embody the parameter that links the brain network structure to its function (i.e., structural and functional matrices). We set $\boldsymbol{\omega} = BD(\boldsymbol{x})\boldsymbol{\delta}$, where $\boldsymbol{x}$ are phase differences obtained from the previous step, and integrate the Kuramoto model in Eq. (1) with random initial conditions close

to $\boldsymbol{x}$. We show in Fig. 8 that the assignment of natural frequencies according to the extracted phase differences promotes spontaneous synchronization to accurately replicate the empirical functional connectivity $F$.

These results corroborate the postulate that structural connections in the brain support the intermittent activation of specific functional patterns during rest through regional metabolic changes. Furthermore, we show that the Kuramoto model represents an accurate and interpretable mapping between the brain anatomical organization and the functional patterns of frequency-synchronized neural co-fluctuations. Once a good mapping is inferred, it can be used to define a generative brain model to replicate in silico how the brain efficiently enacts large-scale integration of information, and to develop personalized intervention schemes for neurological disorders related to abnormal synchronization phenomena[48,49].

**Power flow control in power networks for fault recovery and prevention.** Efficient and robust power delivery in electrical grids is crucial for the correct functioning of this critical infrastructure. Modern, reconfigurable power networks are expected to be resilient to distributed faults and malicious cyber-physical attacks[50] while being able to rapidly adapt to varying load demands. In addition, climate change is straining service reliability, as underscored by recent events such as the Texas grid collapse after Winter Storm Uri in February 2021[51], and the New Orleans blackout after Hurricane Ida in August 2021[52]. Therefore, there exists a dire need to design control methods to efficiently operate these networks and react to unforeseen disruptive events.

The Kuramoto model in Eq. (1) has been shown to be particularly relevant in the modeling of large distribution networks and microgrids[9]. Preliminary work on the control of frequency synchronization in electrical grids modeled through Kuramoto oscillators has been developed in ref. [53]. Here, we present a method that leverages our findings to guarantee not only frequency synchronization but also a target pattern of active power flow. Our method can be used for power (re)distribution with respect to specific pricing strategies, fault prevention (e.g., when a line overheats), and recovery (e.g., after the disconnection

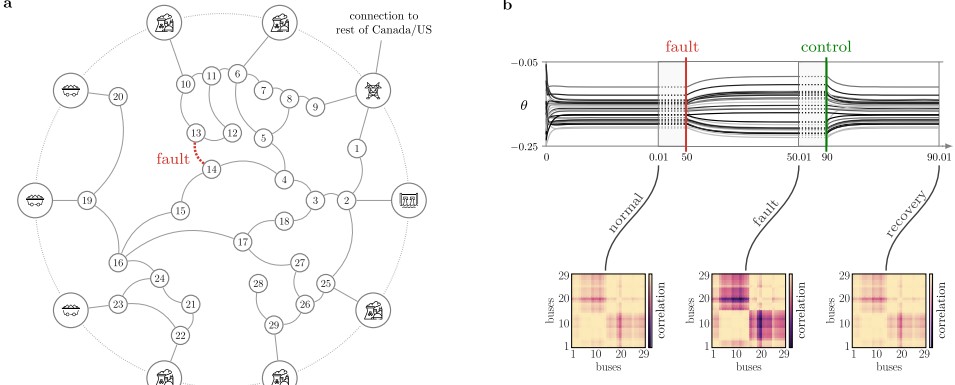

**Fig. 9 Fault recovery in the IEEE 39 New England power distribution network through minimal and local intervention. a** New England power distribution network. The generator terminal buses illustrate the type of generator (coal, nuclear, hydroelectric). We simulate a fault by disconnecting the transmission line 25 (between loads 13 and 14). **b** The fault causes the voltage phases **θ** to depart from normal operating conditions, which could cause overheating of some transmission lines (due to violation of the nominal thermal constraint limits) or abnormal power delivery. To recover the pre-fault active power flow and promote a local (sparse) intervention, we solve the optimization in Eq. (11) by minimizing the 1-norm of the structural parameter modification **δ** with no scaling parameters in the cost functional. The network returns to the initial operative conditions with a localized modification of the neighboring transmission lines' impedances.

of a branch). Furthermore, thanks to the formal guarantees that our method prescribes, we are able to prevent Braess' Paradox in power networks[54], which is a phenomenon where the addition of interconnections to a network may impede its synchronization.

It has been shown in ref. 9, Lemma 1 that the load dynamics (nodes 1–29 in Fig. 9a) in a structure-preserving power grid model have the same stable synchronization manifold of Eq. (1). In this model, $\omega_i = p_{\ell_i}/D_i$ is the active power load at node $i$, where $D_i$ is the damping coefficient, and $A_{ij} = |v_i||v_j|\mathcal{I}(Y_{ij})/D_i$, with $v_i$ denoting the nodal voltage magnitude and $\mathcal{I}(Y_{ij})$ being the imaginary part of the admittance $Y_{ij}$ (see Supplementary Information for details about this model). In this example, we choose a highly damped scenario where $D_i = 1$, which is possibly due to local excitation controllers. Notice that, when the phase angles **θ** are phase-locked and $A_{ij}$ is fixed, the active power flow is given by $A_{ij}\sin(\theta_j - \theta_i)$.

We posit that solving the problem in Eq. (11) to design a local reconfiguration of the network parameters can recover the power distribution before a line fault or provide the smallest parameter changes to steer the load powers to desired values. In practice, control devices such as flexible alternating current transmission systems (FACTs) allow operators and engineers to change the network parameters[55,56]. We demonstrate the effectiveness of our approach by recovering a desired power distribution in the IEEE 39 New England power distribution network after a fault. During a regime of normal operation, we simulate a fault by disconnecting the line between two loads and solve the problem in Eq. (11) to find the minimum modification of the couplings $a_{ij}$ that recovers the nominal power distribution.

We first utilize MATPOWER[57] to solve the power flow problem. Then we use the active powers $\boldsymbol{p}_\ell$ and voltages $v$ at the buses to obtain the natural frequencies **ω** and the adjacency matrix $A$ of the oscillators, respectively, while the voltage phase angles are used as initial conditions $\boldsymbol{\theta}(0)$ for the Kuramoto model in equation (1). We integrate the Kuramoto dynamics and let the voltage phases $\boldsymbol{\theta}(t)$ reach a frequency-synchronized steady state, which corresponds to a normal operating condition. The phase differences also represent a functional pattern across the loads. Next, to simulate a line fault, we disconnect one line. By solving the problem in Eq. (11) with $\boldsymbol{x}_{\text{desired}}$ corresponding to the pre-fault steady-state voltage phase differences, we compute the smallest variation of the

remaining parameters (i.e., admittances) so that the original functional pattern can be recovered. Figure 9b illustrates the effectiveness of our procedure at recovering the nominal pattern of active power flow by means of minimal and localized interventions (see also Supplementary Fig. 8).

The above application is based on a classical lossless structure-preserving power network model[9]. However, in the power systems literature, more complex dynamics that relax some of the modeling assumptions have been proposed. For instance, ref. 58 uses a third-order model (or one-axis model) that takes into account transient voltage dynamics. Instead, ref. 59 studies the case of interconnections with power losses, which lead to a network of phase-lagged Kuramoto–Sakaguchi oscillators[60]. We show in the Supplementary Information that our procedure to recover a target functional pattern can still be applied successfully to a wide range of situations involving these more realistic models.

## Discussion

Distinct configurations of synchrony govern the functioning of oscillatory network systems. This work presents a simple and mathematically grounded mapping between the structural parameters of arbitrary oscillator networks and their components' functional interactions. The tantalizing idea of prescribing patterns in networks of oscillators has been investigated before, yet only partial results had been reported in the literature. Here, we demonstrate that the control of patterns of synchrony can be cast as optimal (convex) design and tuning problems. We also investigate the feasibility of such optimizations in the cases of networks that admit negative coupling weights and networks that are constrained to positive couplings. Our control framework also allows us to prescribe multiple desired equilibria in Kuramoto networks, a problem that is relevant in practice and had not been investigated before.

As stability of a functional pattern may be a compelling property in many applications, we explore conditions to test and enforce the stability of functional patterns. We show that such conditions are rather straightforward in the case of networks that admit both cooperative and competitive interactions. However, stability of target functional patterns in networks that are constrained to only cooperative interactions is a more challenging task, for which we demonstrate that any pattern associated with a

structurally balanced cosine-weighted network cannot be stable. To overcome this issue, we propose a heuristic procedure that adjusts the oscillators' coupling strengths to violate the structurally balanced property and promote the stability of functional patterns with negative correlations. While heuristic, our procedure for stability has proven successful in all our numerical studies. Notice that, differently from methods that study an "average" description of the system at a near-synchronous state (see, e.g., ref. [61]), here we assess the stability of exact target phase-locked trajectories where phase values can instead be arbitrarily spread over the torus. This method can also be extended to assess the stability of equilibria of higher-dimensional oscillators, provided that the considered equilibrium state is a fixed point.

We emphasize that our results are also intimately related to the long-standing economic problem of enhancing network operations while optimizing wiring costs. In any complex system where synchrony between components ensures appropriate functions, it is beneficial to maximize synchronization while minimizing the physical variations of the interconnection weights[62]. Compatible with this principle, neural systems are thought to have evolved to maximize information processing by promoting synchronization through optimal spatial organization[63]. Inspired by the efficiency observed in neurobiological circuitry, Eq.(11) could be utilized for the design of optimal interaction schemes in large-scale computer networks whose performance relies on synchronization-based tasks[64].

An important consideration that highlights the general contributions of the present study is that being able to specify pairwise functional relationships between the oscillators also solves problems such as phase-locking, full, and cluster synchronization. Yet, the converse is not true. In fact, even in the general setting of cluster synchronization[65]—where distinct groups of oscillators behave cohesively—one can only achieve a desired synchronization level *within* the same cluster, but not *across* clusters, which instead is possible with our approach. Specifically, in cluster synchronization, oscillators belonging to the same cluster are forced to synchronize, thus implying that the associated diagonal blocks of the functional pattern $R$ display values close to 1. Seminal work in ref. [66] developed a nonlinear feedback control to change the coupling functions in Eq. (2) to engineer clusters of synchronized oscillators, whereas the authors of ref. [67] propose the formation of clusters through selective addition of interconnections to the network. More recently, the control of partially synchronized states with applications to brain networks is studied in ref. [30] by means of structural interventions, and in ref. [68] via exogenous stimulation. Ultimately, the results presented in this work not only complement but go well beyond the control of macroscopic synchronization observables and partial synchronization by allowing to specify the synchrony level of pairwise interactions.

While our contributions include the analysis of the stability properties of functional patterns, here we do not assess their basins of attraction. We emphasize that, in general, the estimation of the basin of attraction of the attractors of nonlinear systems remains an outstanding problem, and even the most recent results rely on numerical approaches or heavy modeling assumptions[69]. Further, in the case of coupled Kuramoto oscillators, existing literature shows that the number of equilibria for the phase differences increases significantly with the cardinality of the network[36], making the study of basins of attraction extremely challenging. In the Supplementary Information, we extend previous work on identical oscillators to networks with heterogeneous oscillators, and show that functional patterns can be at most almost-globally stable in cluster-synchronized positive networks. Yet, a precise estimation of the basin of attraction for any arbitrary target pattern goes beyond the scope of this work and

may require the derivation of ad-hoc principles based on Lyapunov's stability theory[70].

The framework presented in this work has other limitations, which can be addressed in follow-up studies. First, despite its capabilities in modeling numerous oscillatory network systems, the Kuramoto model cannot capture the amplitude of the oscillations, making it most suitable for oscillator systems where most of the information is conveyed by phase interaction as demonstrated in ref. [40] for resting brain activity. To model brain activity during cognitively demanding tasks such as learning, higher-dimensional oscillators may be more suitable[71]. Second, the use of phase-locked trajectories is instrumental to the control and design of functional patterns. Yet, it is not necessary. In fact, restricting the control to phase-locked dynamics does not capture exotic dynamical regimes in which only a subset of the oscillators is frequency-synchronized. Third and finally, in some situations, the network parameters are not fully known. While still an active area of research, network identification of oscillator systems may be employed in such scenarios[72].

Directions of future research can be both of a theoretical and practical nature. For instance, follow-up studies can focus on the derivation of a general condition for the stability of a feasible functional pattern in positive networks. Further, a thorough investigation of which network structures allow for multiple prescribed equilibria may be particularly relevant in the context of memory systems, where different patterns are associated with different memory states. Specific practical applications may also require the inclusion of sparsity constraints on the accessible structural parameters for the implementation of the proposed control and design framework.

## Methods

**Matrix form of Eq. (1) and phase-locked solutions**. For a given network of oscillators, we let the entries of the (oriented) incidence matrix $B$ be defined component-wise after choosing the orientation of each interconnection $(i, j)$. In particular, $i$ points to $j$ if $i < j$, and $B_{k\ell} = -1$ if oscillator $k$ is the source node of the interconnection $\ell$, $B_{k\ell} = 1$ if oscillator $k$ is the sink node of the interconnection $\ell$, and $B_{k\ell} = 0$ otherwise. The matrix form of Eq. (1) can be written as

$$\dot{\boldsymbol{\theta}} = \begin{bmatrix} \omega_1 \cdots \omega_n \end{bmatrix}^{\mathsf{T}} - B \begin{bmatrix} \ddots & & \\ & \sin(x_{ij}) & \\ & & \ddots \end{bmatrix} \boldsymbol{\delta}$$

$$= \begin{bmatrix} \omega_1 \cdots \omega_n \end{bmatrix}^{\mathsf{T}} - B \operatorname{diag}\Big( \{\sin(x_{ij})\}_{(i,j) \in \mathcal{E}} \Big) \boldsymbol{\delta} = \begin{bmatrix} \omega_1 \cdots \omega_n \end{bmatrix}^{\mathsf{T}} - BD(\boldsymbol{x})\boldsymbol{\delta},$$

where $D(\boldsymbol{x})$ is the diagonal matrix of the sine functions in Eq. (1).

When the oscillators evolve in a phase-locked configuration, the oscillator frequencies become equal to each other and constant. In particular, since $\boldsymbol{1}^{\mathsf{T}}B = 0$, we have $\boldsymbol{1}^{\mathsf{T}}\dot{\boldsymbol{\theta}} = \boldsymbol{1}^{\mathsf{T}}k\boldsymbol{1} = \boldsymbol{1}^{\mathsf{T}}[\omega_1 \cdots \omega_n]^{\mathsf{T}}$, thus showing that in any phase locked trajectory, the oscillator's frequency $k$ needs to equal the mean natural frequency $\frac{1}{n}\sum_{i=1}^{n}\omega_i$.

**Any feasible functional pattern has $n-1$ degrees of freedom**. The values of a functional pattern can be uniquely specified using a set of $n-1$ correlation values. To see this, let us define the incremental variables $\boldsymbol{x} = M\boldsymbol{\theta}$, where $M \in \mathbb{R}^{|\mathcal{E}| \times n}$ is the matrix whose $k$th row, associated to $x_{ij}$, is all zeros except for $b_{ki} = -1$ and $b_{kj} = 1$. Consider the first $n-1$ rows of $M$, associated with $x_{12}, x_{13}, \ldots x_{1n}$, and notice that they are linearly independent. Moreover, the row associated to $x_{ij}$, $i > 1$, can be obtained by subtracting the row associated to $x_{1i}$ to the row associated to $x_{1j}$, implying that the rank of $M$ is $n-1$. Any collection of $n-1$ linearly independent rows of $M$ defines a full row-rank matrix $M_{\min}$ (e.g., any $n-1$ rows corresponding to the transpose incidence matrix of a spanning tree[37]). We let $\boldsymbol{x}_{\min} = M_{\min}\boldsymbol{\theta}$, where $\boldsymbol{x}_{\min}$ is a smallest set of phase differences that can be used to quantify the synchronization angles among all oscillators. Because $\ker(M_{\min}) = \boldsymbol{1}$, we can obtain the phases $\boldsymbol{\theta}$ from $\boldsymbol{x}_{\min}$ modulo rotation: $\boldsymbol{\theta} = M_{\min}^{\dagger}\boldsymbol{x}_{\min} - c\boldsymbol{1}$, where $M_{\min}^{\dagger}$ denotes the Moore–Penrose pseudo-inverse of $M_{\min}$ and $c$ is some real number. Further, since $\ker(M_{\min}) = \ker(M)$, we can reconstruct all phase differences $\boldsymbol{x}$ from $\boldsymbol{x}_{\min}$:

$$MM_{\min}^{\dagger}\boldsymbol{x}_{\min} = M(\boldsymbol{\theta} + c\boldsymbol{1}) = M\boldsymbol{\theta} + 0 = \boldsymbol{x}.$$

The above equation reveals that all the differences $\boldsymbol{x}$ are encoded in $\boldsymbol{x}_{\min}$. That is, any $x_{ij}$ can be written as a linear combination of the elements in $\boldsymbol{x}_{\min}$.

For example, if $n = 3$ and $\boldsymbol{x}_{\min} = [x_{12}\ x_{23}]^{\mathsf{T}}$, then $x_{13}$ is a linear combination of the differences in $\boldsymbol{x}_{\min}$, i.e., $\boldsymbol{x} = \begin{bmatrix} -1 & 1 & 0 \\ 0 & -1 & 1 \\ -1 & 0 & 1 \end{bmatrix} \begin{bmatrix} -1 & 1 & 0 \\ 0 & -1 & 1 \end{bmatrix}^{\dagger} \boldsymbol{x}_{\min}$, in which $x_{13} = x_{12} + x_{23}$. Thus, because $n-1$ incremental variables define all the remaining ones, the entries of any functional pattern must have only $n-1$ degrees of freedom.

**Existence of a strictly positive solution to Problem 5.** Rewrite the pattern assignment problem $BD(\boldsymbol{x})\boldsymbol{\delta} = \boldsymbol{\omega}$ as

$$BD(\boldsymbol{x})\boldsymbol{\delta} = B_{:,\mathcal{H}} D_{\mathcal{H},\mathcal{H}}(\boldsymbol{x})\boldsymbol{\delta}_{\mathcal{H}} + B_{:,\tilde{\mathcal{H}}} D_{\tilde{\mathcal{H}},\tilde{\mathcal{H}}}(\boldsymbol{x})\boldsymbol{\delta}_{\tilde{\mathcal{H}}} = \boldsymbol{\omega},$$

where the subscripts $\mathcal{H}$ and $\tilde{\mathcal{H}}$ denote the entries corresponding to the Hamiltonian path in conditions (ii.a) and (ii.b) and the remaining ones, respectively. Since $\mathrm{Im}(\bar{B}_{:,\mathcal{H}}) = \mathrm{Im}(B_{:,\mathcal{H}} D_{:,\mathcal{H}}(\boldsymbol{x})) = \mathrm{span}(\boldsymbol{1})^{\perp}$, $\mathrm{Im}(B_{:,\tilde{\mathcal{H}}} D_{:,\tilde{\mathcal{H}}}(\boldsymbol{x})) \subseteq \mathrm{span}(\boldsymbol{1})^{\perp}$, and $\boldsymbol{\omega} \in \mathrm{span}(\boldsymbol{1})^{\perp}$, for any vector $\boldsymbol{\delta}_{\tilde{\mathcal{H}}}$, the following set of weights solves the above equation:

$$\boldsymbol{\delta}_{\mathcal{H}} = (B_{:,\mathcal{H}} D_{\mathcal{H},\mathcal{H}}(\boldsymbol{x}))^{\dagger}(\boldsymbol{\omega} - B_{:,\tilde{\mathcal{H}}} D_{\tilde{\mathcal{H}},\tilde{\mathcal{H}}}(\boldsymbol{x})\boldsymbol{\delta}_{\tilde{\mathcal{H}}}) = (D_{\mathcal{H},\mathcal{H}}(\boldsymbol{x}) B_{:,\mathcal{H}}^{\mathsf{T}} B_{:,\mathcal{H}} D_{\mathcal{H},\mathcal{H}}(\boldsymbol{x}))^{-1}$$
$$D_{\mathcal{H},\mathcal{H}}(\boldsymbol{x}) B_{:,\mathcal{H}}^{\mathsf{T}}(\boldsymbol{\omega} - B_{:,\tilde{\mathcal{H}}} D_{\tilde{\mathcal{H}},\tilde{\mathcal{H}}}(\boldsymbol{x})\boldsymbol{\delta}_{\tilde{\mathcal{H}}})$$

Because the matrix $D_{\mathcal{H},\mathcal{H}}(\boldsymbol{x}) B_{:,\mathcal{H}}^{\mathsf{T}} B_{:,\mathcal{H}} D_{\mathcal{H},\mathcal{H}}(\boldsymbol{x})$ is an M-matrix, its inverse has non-negative entries. Further, by condition (ii.b), $D_{\mathcal{H},\mathcal{H}}(\boldsymbol{x}) B_{:,\mathcal{H}}^{\mathsf{T}} \boldsymbol{\omega}$ is strictly positive. Then, the vector $(D_{\mathcal{H},\mathcal{H}}(\boldsymbol{x}) B_{:,\mathcal{H}}^{\mathsf{T}} B_{:,\mathcal{H}} D_{\mathcal{H},\mathcal{H}}(\boldsymbol{x}))^{-1} D_{\mathcal{H},\mathcal{H}}(\boldsymbol{x}) B_{:,\mathcal{H}}^{\mathsf{T}} \boldsymbol{\omega}$ is also strictly positive, and so is the solution vector $\boldsymbol{\delta}_{\mathcal{H}}$ for any sufficiently small and positive vector $\delta_{\tilde{\mathcal{H}}}$.

**Enforcing stability of functional patterns in networks with cooperative and competitive interactions.** To ensure that the Jacobian matrix in Eq. (10) is the Laplacian of a positive network and, thus, stable, we solve the problem in Eq. (5) with a slight modification of the constraints. Specifically, we post-multiply the matrix $B$ in Eq. (5) as $B\Delta$, where $\Delta = \mathrm{diag}(\{\mathrm{sign}(\cos(x_{ij}))\}_{(i,j)\in\mathcal{E}})$ is a matrix that changes the sign of the columns of $B$ associated with negative weights in the cosine-scaled network:

$$B\Delta D(\boldsymbol{x})\boldsymbol{\delta} = \boldsymbol{\omega}$$

Solving for positive interconnection weights the problem in Eq. (5) under the above-modified constraint yields a stable Jacobian in a network where the final couplings are $\Delta\boldsymbol{\delta}$.

**Heuristic procedure to promote stability of functional patterns in positive networks.** We provide a heuristic procedure to promote the stability of functional patterns that include negative correlations in a network with nonnegative weights. Our procedure relies on the definition of Gerschgorin disks and the Gerschgorin Theorem.

**Definition of Gerschgorin disk.** Let $M \in \mathbb{C}^{n\times n}$ be a complex matrix. The $i$th Gerschgorin disk is $\mathcal{D}_i = (M_{ii}, r_i)$, $i = 1, \ldots, n$, where the radius is $r_i = \sum_{j\neq i}|M_{ij}|$ and the center is $M_{ii}$.

**Theorem 2.** (Gerschgorin[39]). *The eigenvalues of the matrix $M$ lie within the union $\bigcup_{i=1}^{n} D_i$ of its Gerschgorin disks.*

Whenever all target phase differences in $\boldsymbol{x}$ satisfy $|x_{ij}| \leq \frac{\pi}{2}$, the Gerschgorin disks of the Jacobian in Eq. (10) all lie in the closed left half-plane. However, for patterns $\boldsymbol{x}$ containing phase differences $|x_{ij}| \geq \frac{\pi}{2}$, the union of the Gerschgorin disks intersects the right half-plane. Reducing the magnitude of the entries satisfying $A_{ij}\cos(x_{ij}) < 0$ effectively shrinks the radius of the Gerschgorin disks that overlap with the right half-plane and shift their centers towards the left-half plane due to the structure of the Jacobian matrix. We remark that the procedure in Eq. (13) is a heuristic, and it is probably effective only when all interconnections with $A_{ij}\cos(x_{ij}) < 0$ can be removed so that all the Gerschgorin disks lie completely in the left-half plane.

## Data availability
The brain data that support the findings of this study are publicly available in the Supplementary Information of[40], at https://doi.org/10.1371/journal.pcbi.1004100.s006. The IEEE 39 New England data parameters and interconnection scheme analyzed in this study for the structure-preserving power network model can be found in the reference textbook[73] (see also Supplementary Information for modeling assumptions). The parameters for the simulations on the network-reduced IEEE 39 New England test case in the Supplementary Information have been obtained from refs. [74,75]. All data used in this study are also available in the public GitHub repository: https://github.com/tommasomenara/functional_control.

## Code availability
Source code and documentation for the numerical simulations presented here are freely available in GitHub at: https://github.com/tommasomenara/functional_control with the identifier https://doi.org/10.5281/zenodo.4546413.

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

## Acknowledgements

T.M. was supported in part by the National Science Foundation (Grant NSF-557 NCS-FO-1926829) and the Army Research Office (Grant ARO-W911NF1910360).

## Author contributions

T.M., G.B., and F.P. designed research; T.M. and G.B. performed research; T.M. analyzed data and developed code; D.S.B and F.P. supervised the research, and T.M., G.B., D.S.B, and F.P. wrote the paper.

## Competing interests

The authors declare no competing interests.
