## [Peer Review File · Nature Communications]

REVIEWER COMMENTS

Reviewer #1 (Remarks to the Author):

In the manuscript, the authors tackle the famous question of how structural parameters (coupling structure, heterogeneous local node dynamics) of a dynamical system relate to the system's functional properties. In order to get closer to an answer to this fundamental question, a system of coupled heterogeneous Kuramoto oscillators is analyzed.

By applying methods from algebraic graph theory, the authors derive sufficient conditions that guarantee the existence of a structural configuration (either given by a certain weighted coupling structure and/or distribution of natural frequencies) such that the corresponding Kuramoto system possesses a phase-locked solution that replicates a prescribed functional pattern. Thus, a map from a given functional pattern to a set of structural parameters is achieved for coupled phase oscillators. The existence conditions are expressed in complete algebraic and rather graph-theoretic terms. All results are very well explained, the derivations are presented in a clear fashion and an elucidative example is provided.

In order to determine computationally an actual structure (network and/or frequency) for one or even several prescribed functional patterns, a convex optimization problem is formulated that can be solved by using standard methods. An example is provided illustrating the results obtained by solving the optimization problem for two simultaneously prescribed functional patterns. At this point, I would like to highlight the strength of this methodology by mentioning that the co-existence of phase-locked solutions is an omnipresent feature in many complex dynamical systems. Thus, knowing which structural configuration may foster certain locked patterns is important for applied (e.g. to study working memory) and theoretical (e.g. to study explosive synchronization phenomena) research.

In addition to the existence criteria, the authors analyze the local exponential stability properties for a given functional pattern. A sufficient condition for the instability of functional patterns is given. Building on this, an intriguing extension of the previously stated optimization problem is presented by separating positive and negative entries of the generally signed Laplacian matrix which corresponds to the local stability problem. Even though the computational method is interesting, it remains unclear to me how stability can be guaranteed via this method as it relies on a sufficient condition for instability rather than for stability. In my opinion, this issue has to be addressed in a revision of the manuscript.

Further, two applications of the developed methodology are provided. The first example concerns dynamical brain networks and the second, the control of power grid systems in the case of a line failure. The problems that are addressed by the authors by applying their method to the example systems are important for the fields of neuroscience and engineering.

In the last part of the manuscript, limitations of the presented approach are well described and interesting future research perspectives are provided. The method developed in this manuscript provides a new tool to study many frequency-locking-related questions in real-world systems. I believe the method will be of interest to researchers from various fields that are interested in the existence of phase-locked states that replicate a given functional pattern. With regards to the stability of the patterns, in my opinion, more has to be done (see below).

In the following, I list a couple of remarks and concerns on the generality of the developed approach.

(1) Already in the title the word "control" is prominently placed. However, for me, it is

not clear in which sense the functional patterns are controlled? In addition, to guarantee the existence of a certain pattern and its local stability, an effective control scheme should also guarantee that a given dynamical system achieves the desired output. For this, I think more details on the multistability of patterns are needed. Further, one could ask the question: Is it possible to invoke conditions that guarantee global stability of a pattern? Would it be possible to quantify the basins of attraction for certain patterns? Furthermore, in the dynamical brain network example, how likely is it to obtain the prescribed functional pattern from random initial conditions; are there other co-existing stable patterns?

(2) With regards to stability, I believe the extended optimization problem is the main result as it provides the novelty of the author's stability analysis. However, the authors should comment on the fact that a solution (structural configuration) of the proposed optimization process does not imply stability of the corresponding phase-locked state.

(3) As outlined in the manuscript, phase-locked states are very important collective states in systems of coupled oscillators. I support the author's assumptions that the Kuramoto model is a good approximation for dynamical systems close to frequency synchronization. In the case of a brain's resting state, the authors show that the phase-locked pattern resembles very well the functional pattern from experimental data. As also discussed by the authors, the presented methodology is mathematically restricted to phase oscillator models. Therefore, with regards to applications to more complex (e.g. higher dimensional) dynamical systems, a dynamically close relation to the Kuramoto system is needed as e.g. given in the case of resting states. However, the Kuramoto system would only be a good approximation for the existence of certain phase-locked states. The stability, in contrast, is not necessarily well approximated by the Kuramoto model and can depend on higher-dimensional dynamical features of the individual nodes. For instance in Sun et.al., Europhys. Lett. 85, 6, 60011 (2009) the complex local dynamics enters explicitly the master stability equation for nearly synchronized oscillators. There might be "stable" functional patterns extracted from experimental data (from a system with complex high dimensional node dynamics) for which the proposed method might not find a structural configuration such that the pattern is stable in the Kuramoto dynamics as well. In this case, the additional complex features would have stabilized the state. In turn, there might be also co-existing stable patterns in the Kuramoto dynamics for a given structural configuration that wouldn't be stable in the more complex system. Here, complex features in the local dynamics might destabilize certain patterns.

(4) With regards to the example on the control of power grids, I am concerned whether the developed methodology also applies to more complex power system models such as analyzed in Sharafutdinov et.al., Chaos 28, 3, 033117 (2018) or Hellmann et. al., Nat. Commun. 11, 592 (2020). I believe that most of the ideas developed in this work could still be applied. However, nontrivial modifications to the methodology are needed to render it applicable for other systems.

Minor remarks and comments:

(i) The adjacency matrix is assumed to be symmetric which is another restriction of the formalism. Is it possible to lift this restriction?

(ii) How are the parameters c_1 and c_2 chosen in the power grid example?

(iii) I think (i.b) implies (i.c). Please clarify.

(iv) Please add some details, how the last equation of page 6 is derived. It took me some minutes to get what is meant by the expressions in the equation.

(v) Is there a bound on how many patterns can be prescribed in the "multi-pattern" optimization problem for a given system with n oscillators?

(vi) In power grids the natural frequency is given by $\omega = p/D$ where D is the damping constant.

(vii) Suppl. Mat.: In Thm 1, it should read $[\gamma, \phi_2, \dots, \phi_{(n-1)}]$. In the proof, do you distinguish between n odd and n even?

(viii) Suppl. Mat.: There are two zeros in front of the dot in Eq.(8), please drop one zero.

Reviewer #2 (Remarks to the Author):

This paper studies the control of functional patterns in oscillatory networks. The work is based on considering phase models, and the main contribution is to develop principled method to enforce desired functional patterns by optimal tuning of oscillator parameters.

Overall, the paper is well structured, and the problem of functional control is relevant and important to the field of oscillatory networks. However, the overall contribution of the work is very limited. The control of functional patterns, e.g., phase-locking, synchronization, or cluster patterns, in oscillatory networks has been extensively studied. In particular, existing works on optimal control and optimization for phase assignment using phase-models can be widely found in the literature. In addition, the proposed formulations and techniques for functional/pattern control by tuning network weights and natural frequencies of oscillators are quite standard in control systems. One unique point addressed in this work is the consideration of positive couplings, which is practically relevant, however, it can be easily accommodated as a constraint in the formulated optimization problem as done in the paper.

The paper is mostly technically sound, and the content is much more suitable to a specialized domain journal, e.g., in control systems or nonlinear science. The proposed analytical results, including the optimal control and convex optimization formulations and approaches, as well as the stability analysis are within the domain of classical control systems. Also, the presented applications to complex networks are limited to Kuramoto oscillators.

Here are some specific comments:

1. As the paper has a strong technical flavor, there is no rigorous proof of the conditions that guarantee the existence of the solution of the linear inverse problem. Especially, on Page 6, it is incorrect to argue the existence of a positive solution by saying merely that an arbitrary small solution can be found.

2. On Page 7, in the optimization problem in (7), which is an extension of the one in (6) on the same page, why is the positivity constraint dropped?

3. As shown in the example in Fig. 1, the corrected network has fewer edges than the original one, and hence during the phase assigning procedure, the rank of the matrix B in (7) may be decreased, does this impact on the feasibility of the proposed method?

4. In the proposed method, it is crucial to find a Hamiltonian path that represents the solution. Does every Hamiltonian path generate the same result? If not, then how to apply the proposed method, e.g., in the case that the network is sparsely connected such that the desired Hamiltonian path is not in the network.

5. In the application to brain networks, the authors claim that Kuramoto model gives a good approximate to the rhythmic activity of brain networks. However, there is no verification that the fMRI data fit the model well so that the results obtained by the

proposed method is convincing.

6. The authors mention that the concept of functional pattern proposed in this work is a good replacement of the classical Pearson correlation. What are the benefits of using functional pattern instead of the Pearson correlation? This should be justified.

7. The section "Assessing and enforcing stability of functional patterns" contains insufficient details to understand the proposed heuristic. It will be helpful if the steps can be illustrated.

Response to Reviewer 1

We thank the reviewer for their constructive comments. Note that all the changes following this reviewer's comments are highlighted in blue in the revised manuscript.

[R1: 1]

In the manuscript, the authors tackle the famous question of how structural parameters (coupling structure, heterogeneous local node dynamics) of a dynamical system relate to the system's functional properties. In order to get closer to an answer to this fundamental question, a system of coupled heterogeneous Kuramoto oscillators is analyzed.

By applying methods from algebraic graph theory, the authors derive sufficient conditions that guarantee the existence of a structural configuration (either given by a certain weighted coupling structure and/or distribution of natural frequencies) such that the corresponding Kuramoto system possesses a phase-locked solution that replicates a prescribed functional pattern. Thus, a map from a given functional pattern to a set of structural parameters is achieved for coupled phase oscillators. The existence conditions are expressed in complete algebraic and rather graph-theoretic terms. All results are very well explained, the derivations are presented in a clear fashion and an elucidative example is provided.

In order to determine computationally an actual structure (network and/or frequency) for one or even several prescribed functional patterns, a convex optimization problem is formulated that can be solved by using standard methods. An example is provided illustrating the results obtained by solving the optimization problem for two simultaneously prescribed functional patterns. At this point, I would like to highlight the strength of this methodology by mentioning that the co-existence of phase-locked solutions is an omnipresent feature in many complex dynamical systems. Thus, knowing which structural configuration may foster certain locked patterns is important for applied (e.g. to study working memory) and theoretical (e.g. to study explosive synchronization phenomena) research.

Further, two applications of the developed methodology are provided. The first example concerns dynamical brain networks and the second, the control of power grid systems in the case of a line failure. The problems that are addressed by the authors by applying their method to the example systems are important for the fields of neuroscience and engineering.

In the last part of the manuscript, limitations of the presented approach are well described and interesting future research perspectives are provided. The method developed in this manuscript provides a new tool to study many frequency-locking-related questions in real-world systems. I believe the method will be of interest to researchers from various fields that are interested in the existence of phase-locked states that replicate a given functional pattern.

Response: We thank the reviewer for the positive evaluation of our work and their detailed assessment of our manuscript.

For this reviewer's convenience, we list below the main additions and changes to the manuscript that regard this reviewer's suggestions.

The major additions to the manuscript as follows.

- (a) We have extended our optimization method to include the case of asymmetric adjacency matrices and, thus, directed networks;
- (b) We have shown that, in general, there does not exist a bound on the number of functional patterns that can be concurrently assigned in a network;
- (c) We have shown that the kernel of the incidence matrix uniquely determines all compatible functional patterns of a network;
- (d) We have applied our procedure to alternative and more faithful models of power networks;
- (e) We have investigated the basin of attraction of certain classes of functional patterns. We have shown that, in the case of cluster-synchronized positive networks, functional patterns associated with phase-locked trajectories are at most almost-globally stable.

We have also rewritten the part “*Analytical Results*”. Besides addressing this reviewer’s comments, this part now contains novel results (particularly with respect to the coexistence of multiple functional patterns), a simplified notation, a clearer exposition, and more examples. With respect to the reviewers’ comments, the major changes we made in the revised version are as follows.

- (a) We have enhanced the text in the section “*Feasible functional patterns in positive networks*” that explains why conditions (ii.a) and (ii.b) are sufficient for the existence of a strictly positive solution. We have also added a section in the Methods containing the technical details used to demonstrate our claim;
- (b) We have expanded the derivation of the equations concerning the feasibility of desired functional patterns in strictly positive networks possessing Hamiltonian paths;
- (c) We have extended and enhanced our treatment of the stability of functional patterns. The main manuscript contains now a more comprehensive description and an application of our heuristic method to promote stability of functional patterns with negative correlations in positive networks.

[R1: 2] *With regards to the stability of the patterns, in my opinion, more has to be done (see below).*

Response: Thank you for pointing out an important research direction. We summarize below the additional analyses and results that we have added to the paper. While we could not completely characterize the basin of attraction of arbitrary functional patterns – as it remains a challenging research problem that very few works have only partially tackled – we were able to derive some partial results that we discuss more in detail in [R1:4]. In a nutshell, in [R1:4] we show that, in the case of positive networks, functional patterns associated with cluster-synchronized trajectories are at most almost-globally stable. Further, for phase-unlocked trajectories (i.e., oscillators evolving with different frequencies) numerical evidence suggests that global stability of specific patterns can be achieved.

[R1: 3] *Already in the title the word “control” is prominently placed. However, for me, it is not clear in which sense the functional patterns are controlled?*

Response: Our method to compute structural parameters that achieve desired functional patterns can be used in (offline) network design tasks and (online) intervention problems. In contrast to classical control schemes, we do not require the oscillators’ phases to be measurable, as our procedure relies on the system’s parameters being the control knobs to achieve the desired functional pattern. Thus, with our method, functional patterns can effectively be *controlled* by tuning the network weights and/or the oscillators’ natural frequencies.

For the above reason, we would prefer to keep the word *control*, since it explicitly encompasses the main scope of our work. However, since we agree that here we do not develop a control scheme in its “classical” sense, we would be happy to consider any suggestion that this reviewer might have in regards to the title.

[R1: 4] *In addition, to guarantee the existence of a certain pattern and its local stability, an effective control scheme should also guarantee that a given dynamical system achieves the desired output. For this, I think more details on the multistability of patterns are needed. Further, one could ask the question: Is it possible to invoke conditions that guarantee global stability of a pattern? Would it be possible to quantify the basins of attraction for certain patterns?*

Response: Thank you for this comment and the great suggestions. We remark that, in general, the analysis and estimation of the basin of attraction of nonlinear systems remains an outstanding problem, and even the most recent results rely on numerical approaches or heavy modeling assumptions [1]. To the best of our knowledge, Ref. [2] provides the most up-to-date study of the basins of attraction of synchronized Kuramoto oscillators. However, the authors only derive estimates of the basin of attraction for the fully synchronized case in networks of identical oscillators. While a complete characterization of a general pattern’s basin of attraction remains an open challenge, below we provide new analyses on multistability and the patterns’

basins of attractions for the case of cluster-synchronized positive networks. We find that functional patterns associated with phase-locked trajectories in this large class of networks are at most almost-globally stable. Our results extend previous work on identical oscillators to the case of oscillators with heterogeneous natural frequencies.

Existing literature shows that the number of equilibria for the phase differences of heterogeneous oscillators evolving on random graphs increases significantly with the cardinality of the network [3]. Therefore, we begin by restricting our analysis to the case of identical oscillators. In general, for any connected network of identical oscillators with cooperative connections $A_{ij} \geq 0$, there is no unique pattern. In fact, the largest basin of attraction is achieved by \mathbb{S}^1 -synchronizing graphs (e.g., complete graphs, acyclic graphs, and sufficiently dense graphs [4, 5]), which feature almost-global stability of the fully synchronized functional pattern. In this class of networks, the only stable equilibrium $\mathbf{x}_{\text{desired}}$ satisfies $x_{ij} = 0$ for all i, j , and there exists a finite number of unstable equilibria. For instance, consider two connected identical oscillators: $\omega_1 = \omega_2$. The only stable equilibrium for the phase differences dynamics is $x_{12} = \theta_2 - \theta_1 = 0$. Yet, there also exists an unstable equilibrium $x_{12} = \pi$, making the equilibrium $x_{12} = 0$ only almost-globally stable. Note that, in general, almost-global stability cannot be guaranteed for arbitrary classes of sparse networks of identical oscillators, as other synchronization manifolds besides the fully synchronized one can be stable – for example, splay states emerge in Cayley graphs [6]. This latter class of graphs admits two stable equilibria (full synchronization and splay states), which implies the coexistence of two stable patterns.

We can generalize the above observations to networks of heterogeneous oscillators. To do so, we leverage cluster synchronization – a phenomenon where distinct groups of synchronized oscillators coexist in a network [7] – of \mathbb{S}^1 -synchronizing clusters with possibly different natural frequencies. We consider cluster synchronization in networks with a partition of the oscillators $\mathcal{C} = \{\mathcal{C}_1, \dots, \mathcal{C}_m\}$, with $\mathcal{C}_k \subseteq \mathcal{O}$ being a subset of the network oscillators constituting an \mathbb{S}^1 -synchronizing graph, $k \in \{1, \dots, m\}$, where $\bigcup_{k=1}^m \mathcal{C}_k = \mathcal{O}$ and $\mathcal{C}_k \cap \mathcal{C}_\ell = \emptyset$ if $k \neq \ell$. Further, $\omega_i = \omega_j$ for every $i, j \in \mathcal{C}_k$, $k \in \{1, \dots, m\}$.¹ Whenever cluster synchronization emerges, the diagonal blocks (of sizes $|\mathcal{C}_k| \times |\mathcal{C}_k|$, $k = 1, \dots, m$) of the functional pattern associated with partition \mathcal{C} satisfy $\rho_{ij} = 1$ (see Fig. 1).² Thus, for a given number of clusters $m \geq 1$, each of the $\binom{n}{m}$ possible ways to partition the network yields a functional pattern with diagonal blocks corresponding to synchronized clusters.

Figure 1: **A network of $n = 12$ oscillators with partition $\mathcal{C} = \{\mathcal{C}_1, \dots, \mathcal{C}_4\}$ and the functional pattern R associated with cluster-synchronized trajectories.** Each cluster consists of synchronized oscillators, which produce a correlated diagonal block in the pattern R that satisfy $\rho_{ij} = 1$ for all $i, j \in \mathcal{C}_k$, $k \in \{1, 2, 3, 4\}$.

By extending the above observations on the stability of \mathbb{S}^1 -synchronizing graphs, we find that, in networks where at least one cluster satisfies $|\mathcal{C}_k| \geq 2$, at most almost-global stability of cluster-synchronized functional patterns can be achieved. In fact, for any choice of intra-cluster phase differences from the finite set of unstable equilibria in \mathbb{S}^1 -synchronizing graphs (i.e., $x_{ij} = \pi$ for at least one intra-cluster phase difference), there exists

¹This is a necessary and sufficient condition for cluster synchronization [7].

²Without loss of generality, we assume that the network oscillators are labeled such that consecutive labels belong to the same cluster.

a value for the inter-cluster phase differences for which the network admits phase-locked trajectories where intra-cluster phase differences satisfy $x_{ij} = 0$ or $x_{ij} = \pi$.

As an example, consider a 4-oscillator network partitioned as $\mathcal{C} = \{\mathcal{C}_1, \mathcal{C}_2\}$, where $\mathcal{C}_1 = \{1, 2\}$ and $\mathcal{C}_2 = \{3, 4\}$. The network parameters read

$$A = \begin{bmatrix} 0 & 5 & 2 & 0 \\ 5 & 0 & 0 & 2 \\ 2 & 0 & 0 & 6 \\ 0 & 2 & 6 & 0 \end{bmatrix} \quad \text{and} \quad \boldsymbol{\omega} = \begin{bmatrix} 1 \\ 1 \\ 2 \\ 2 \end{bmatrix}.$$

Note that the two clusters are \mathbb{S}^1 -synchronizing graphs. It can be shown that there exists an unstable equilibrium at $\boldsymbol{x}_{\text{desired}} = [x_{12} \ x_{23} \ x_{34}]^T = [\pi \ 0.25268 \ \pi]^T$. Thus, whenever the pattern R associated to the partition \mathcal{C} is stable, it is at most almost-globally stable.

We conclude this analysis by investigating the stability of cluster-synchronized trajectories in networks where clusters are not \mathbb{S}^1 -synchronizing. While this paper focuses on phase-locked trajectories, numerical evidence suggests that if the natural frequencies across different clusters are sufficiently heterogeneous (i.e., $\boldsymbol{x}_{\text{inter}}$ is not phase-locked and the clusters evolve at different frequencies), then the functional pattern associated to cluster-synchronized trajectories is almost-globally stable. A further investigation of this dynamical regime goes beyond the scope of this work, and we leave the characterization of the basin of attraction of patterns associated to phase-unlocked trajectories as a topic for future research.

We have added the above discussion to the Supplementary Information, and we now briefly mention in the section *Contribution and Future Directions* of the main text that functional patterns possess at most almost-global stability in cluster-synchronized positive networks:

“While our contributions include the analysis of the stability properties of functional patterns, here we do not assess their basins of attraction. We emphasize that, in general, the estimation of the basin of attraction of nonlinear systems remains an outstanding problem, and even the most recent results rely on numerical approaches or heavy modeling assumptions [1]. Further, in the case of coupled Kuramoto oscillators, existing literature shows that the number of equilibria for the phase differences of oscillators increases significantly with the cardinality of the network [3], making the study of basins of attraction extremely challenging. In the Supplementary Information, we extend previous work on identical oscillators to networks with heterogeneous oscillators, and show that that functional patterns can be at most almost-globally stable in cluster-synchronized positive networks. Yet, a precise estimation of the basin of attraction for any arbitrary target pattern goes beyond the scope of this work and may require the derivation of *ad-hoc* principles based on Lyapunov’s stability theory [2]. ”

[R1: 5] *Furthermore, in the dynamical brain network example, how likely is it to obtain the prescribed functional pattern from random initial conditions; are there other co-existing stable patterns?*

Response: We numerically investigate the size of the basin of attraction of the prescribed functional connectivity pattern, and find that any trajectory starting within an arc distance of $\frac{\pi}{2}$ from the desired phase differences converges to the same pattern extracted from the functional MRI recordings. Fig. 2 illustrates the convergence of 10^4 random initializations of the phases in this region of the torus.

Our numerical simulations reveal that phase trajectories starting from the other half of the torus (i.e., $\|\boldsymbol{x}(0) - \boldsymbol{x}\|_{\infty} > \frac{\pi}{2}$) may converge to different stable patterns, which are not compatible with the phases extracted from the functional MRI recordings.

We have briefly commented on the stability of the functional pattern on the main text and added this figure to the Supplementary Information.

[R1: 6] *With regards to stability, I believe the extended optimization problem is the main result as it provides the novelty of the author’s stability analysis. However, the authors should comment on the fact that a solution (structural configuration) of the proposed optimization process does not imply stability of the corresponding phase-locked state.*

Figure 2: **Stability of the prescribed functional pattern from random initial conditions x_0 satisfying $\|x_0 - x\|_\infty \leq \frac{\pi}{2}$.** The thick black line represents the average ℓ_2 -norm distance over 10^4 random initializations, and the shaded area represents the smallest and largest value of the ℓ_2 -norm distance. The sudden drop in the largest norm is due to the phases $\theta \in \mathbb{T}^n$ evolving in the torus, thus taking values in the interval $[0, 2\pi)$.

Response: Following the reviewer’s suggestion, we have added a paragraph to discuss the heuristic procedure more in detail in the section *Contribution and future directions* of the main text:

“Because stability of a functional pattern may be a compelling property in many applications, we explore conditions to test and prescribe functional patterns’ stability. We show that such conditions are rather straightforward in the case of networks that admit both cooperative and competitive interactions. However, assessing and enforcing stability of target functional patterns in networks that are constrained to only cooperative interactions is a more challenging task. In this case, we demonstrate that any pattern associated to a structurally-balanced cosine-weighted network cannot be stable. To overcome this issue, we propose an heuristic procedure that adjusts the oscillators’ coupling strengths to promote the stability of functional patterns with negative correlations. While there might exist solutions of our procedure that are not able to stabilize the corresponding phase-locked state, our procedure provided successful results in all our attempts.”

We have also enhanced the description of the heuristic procedure in the main text and moved useful results from the Supplementary Information to the Methods section. Moreover, we have added a figure in the main text that shows how the Gerschgorin disks and the Jacobian’s eigenvalues shift towards the (stable) left-half plane for the network in the example of Supplementary Text 1.5 (see also Fig. 10 below)

[R1: 7] *As outlined in the manuscript, phase-locked states are very important collective states in systems of coupled oscillators. I support the author’s assumptions that the Kuramoto model is a good approximation for dynamical systems close to frequency synchronization. In the case of a brain’s resting state, the authors show that the phase-locked pattern resembles very well the functional pattern from experimental data. As also discussed by the authors, the presented methodology is mathematically restricted to phase oscillator models. Therefore, with regards to applications to more complex (e.g. higher dimensional) dynamical systems, a dynamically close relation to the Kuramoto system is needed as e.g. given in the case of resting states. However, the Kuramoto system would only be a good approximation for the existence of certain phase-locked states. The stability, in contrast, is not necessarily well approximated by the Kuramoto model and can depend on higher-dimensional dynamical features of the individual nodes. For instance in Sun et.al., Europhys. Lett. 85, 6, 60011 (2009) the complex local dynamics enters explicitly the master stability equation for nearly synchronized oscillators. There might be “stable” functional patterns extracted from experimental data (from a system with complex high dimensional node dynamics) for which the proposed method might not find a structural configuration such that the pattern is stable in the Kuramoto dynamics as well. In this case, the additional complex features would have stabilized the state. In turn, there might be also co-existing stable patterns in the Kuramoto dynamics for a given structural configuration that wouldn’t be stable in the more complex system. Here, complex features in the local dynamics might destabilize certain patterns.*

Response: We agree with the reviewer that our results are based on first-order Kuramoto dynamics being a good approximation of the dynamical features of a system. As the reviewer points out this choice may not be restrictive – Kuramoto oscillators have been shown to be good approximations of a wide range of systems in multiple domains. Examples include power systems, Josephson junctions, neuronal networks, pacemaker cells in the heart, circadian cells, coupled analog clocks, and flocking, schooling, and vehicle coordination (see, e.g., the surveys [8, 4] and references therein).

Differently from methods that compute an “average” description of the system at near-synchronous state, such as the extended Master Stability Function approach proposed in Ref. [9], we leverage the Jacobian of the phase dynamics at the exact target phase-locked trajectories (whose values can be arbitrarily spread over the torus). Thus, we argue that the stability methods proposed in the paper can be used for systems that do not explicitly manifest complex higher-dimensional dynamical behaviors around the desired operating regime. As we show in [R1:8] below, an example of this behavior is when the complex voltage dynamics in a third-order power network model do not display significant variance or swings around the working point.

Nevertheless, for oscillator systems with nontrivial high-dimensional local features such as complex amplitude dynamics intertwined with the phases state, stability of functional patterns may not be assessed simply by looking at the eigenvalues of the Laplacian of the oscillators’ phases. As the reviewer points out, dynamical features of the individual nodes may require fine-tuned methods to assess the stability of a target pattern. One way to explicitly account for higher-dimensional dynamics is to assess stability of the Jacobian matrix computed at an equilibrium that comprises target values for the entire state of the system (e.g., for phase *and* amplitude dynamics in 2-dimensional oscillators). This approach, however, provides information on the stability of a specific equilibrium of the system only in the vicinity of such equilibrium. A different approach that overcomes this limitation is to resort to Lyapunov’s direct method [10]. Differently from Jacobian analysis, this method requires the crafting of a system-dependent Lyapunov function, and typically cannot be generalized.

We have summarized the above discussion in the section “Contribution and future directions”:

“Notice that, differently from methods that study an “average” description of the system at near-synchronous state (see, e.g., Ref. [9]), here we assess the stability of exact target phase-locked trajectories where phase values can be arbitrarily spread over the torus. This method can also be extended to assess the stability of equilibria of higher-dimensional oscillators, provided that the considered equilibrium state is a fixed point.”

[R1: 8] *With regards to the example on the control of power grids, I am concerned whether the developed methodology also applies to more complex power system models such as analyzed in Sharafutdinov et.al., Chaos 28, 3, 033117 (2018) or Hellmann et. al., Nat. Commun. 11, 592 (2020). I believe that most of the ideas developed in this work could still be applied. However, nontrivial modifications to the methodology are needed to render it applicable for other systems.*

Response: Thank you for suggesting more realistic power grid models to test our methodology. In what follows, we show that our procedure can still be applied in a large class of cases for the models in Sharafutdinov *et al.*, Chaos 28 (here, Ref. [11]) and in Hellmann *et al.*, Nat. Commun. 11, 592 (2020) (here, Ref. [12]). Moreover, in the latter case, we develop a modification to our procedure that explicitly accounts for the phase shift caused by energy losses.

Ref. [11] utilizes a third-order (or one-axis) model. The main differences with the structure-preserving power network model that we utilized in our application is that the one in Ref. [11] includes the transient dynamics of voltage magnitudes, and that electrical loads are simply modeled as passive impedances. For $N = 10$ generators, Fig. 3a illustrates the reduced power network model of the IEEE 39 test case obeying the dynamics [11]

$$\begin{cases} \dot{\theta}_i = \omega_i, \\ M_i \dot{\omega}_i = p_{m,i} - D_i \omega_i + \sum_{j=1}^N |v_i| |v_j| \text{Im}(Y_{ij}) \sin(\theta_j - \theta_i), \\ T_i \dot{v}_i = v_i^f - v_i + (\chi'_i - \chi_i) \sum_{j=1}^N |v_j| \text{Im}(Y_{ij}) \cos(\theta_j - \theta_i), \end{cases} \quad (1)$$

where θ_i is the rotor angle, ω_i its frequency, $p_{m,i}$ is the effective mechanical input power of the machine i , M_i and D_i are the inertia and damping of the mechanical motion, respectively, v_i indicates the transient voltage, $\text{Im}(Y_{ij})$ is the susceptance of the transmission line (i, j) , T_i denotes the relaxation time of the transient voltage dynamics along the q axis, v_i^f is the internal voltage, and, finally, χ_i and $x = \chi'_i$ are the static and transient reactances along the d -axis.

Figure 3: **Application of our procedure to the third-order model of the IEEE 39 New England.** **a** The IEEE-39 New England power network reduced to a 10-generator network. Electrical loads are simply modeled as passive impedances. In order to explicitly account for the outside of the system, Generator 1 is assumed to be connected to an infinite bus and has constant phase and frequency [13]. **b** The absolute difference between the pre-fault functional pattern R_0 and the post-fault pattern R_{fault} . The Frobenius norm of this difference is $\|R_0 - R_{\text{fault}}\|_F = 0.0907$. **c** The absolute difference between the pre-fault functional pattern R_0 and the recovered pattern $R_{\text{recovered}}$ after tuning the power \mathbf{p}_m at the generators. The Frobenius norm of this difference is $\|R_0 - R_{\text{recovered}}\|_F = 0.0653$.

Akin to the structure-preserving power network model, stationary operation of the grid corresponds to constant voltages and frequencies, along with constant rotor phase differences. Since our procedure relies on phase-locked trajectories to achieve a desired functional pattern (i.e., power flow), it can be adjusted to intervene

on the stationary operation of a challenging model such as the one in equations (1). In fact, in regimes with small voltage and frequency swings, the system is still well approximated by first-order Kuramoto dynamics. Clearly, for a desired functional pattern, the error between the one estimated from the application of our procedure and the one obtained from the dynamics in equations (1) will be proportional to the changes in voltages and frequencies.

To test our approach on the model from equations (1), we use the same IEEE 39 New England benchmark case as in the main manuscript. The initial network parameters for our simulations, which represent standard grid operating conditions, are taken from Ref. [13] and Ref. [14]. The voltage $v_i(0)$ and the initial condition for $\theta_i(0)$ and $\omega_i(0) = 0$ for generator i are fixed using power flow computation. The goal of this application is to recover the initial power flow after the same fault as in Ref. [13] (a line trip between loads 16 and 17) occurs. Such a fault affects the network admittance matrix, and yields an undesired power flow. To recover the pre-fault power flow we follow the same steps as described in the main text (see also Fig. 5b of the manuscript), but because the reduced-network structure is typically a complete graph [15], we do not modify the coupling strengths. Instead, we assume that we can adjust the values of $\mathbf{p}_m \in \mathbb{R}^N$, so that $\mathbf{p}_m - \text{diag}(D_1, \dots, D_N)\boldsymbol{\omega}$ are the oscillators' natural frequencies that are tuned in order to achieve the pre-fault functional patterns in the post-fault network. The values for P_m are computed by rewriting the second one of equations (1) as equation (4) in the main text, and solving for the natural frequencies. For the coupling values in $\boldsymbol{\delta}$, we use $a_{ij} = |v_i^{\text{ss}}||v_j^{\text{ss}}|\text{Im}(Y_{ij})$, where v_i^{ss} denotes the steady state value of the voltage v_i before applying our intervention. Finally, we sum a positive constant to the obtained natural frequencies so that they are all positive – this constant can be used to adjust the generators to a desired average mechanical input.

Fig. 3b-c illustrate that our procedure is able to adequately recover the desired functional pattern after the line trips occurs. The error between the pre-fault pattern R_0 and the recovered one $R_{\text{recovered}}$ is due to small changes in the frequency and voltage values after P_m is modified. We emphasize that our procedure relies on the dynamics of $\boldsymbol{\omega}$ and \mathbf{v} leading to small changes. In situations where these changes significantly affect the phases dynamics, the first-order Kuramoto approximation leveraged by our method may not successfully recover a desired functional pattern.

The second reference mentioned by the reviewer (Ref. [12]) uses a power network model where physical losses are not neglected. By compensating for the losses in the injected power but considering lossy interconnections, the structure-preserving model in equations (9) of the Supplementary Information becomes,

$$\begin{cases} M_i \ddot{\theta}_i + D_i \dot{\theta}_i &= \omega_i + \sum_{j=1}^{|\mathcal{V}_1|} A_{ij} \sin(\theta_j - \theta_i + \varphi), \quad i \in \mathcal{V}_1, \\ D_i \dot{\theta}_i &= \omega_i + \sum_{j=1}^{|\mathcal{V}_2|} A_{ij} \sin(\theta_j - \theta_i + \varphi), \quad i \in \mathcal{V}_2, \end{cases} \quad (2)$$

where $\varphi \in \mathbb{S}^1$ represents the phase shift induced by energy losses. We show in Fig. 4 that our procedure to restore a desired functional pattern in a lossy power network after a fault still works well for small values of the phase shift φ .

At synchronous operating conditions, the dynamics in equations (2) reduce to the ones of Kuramoto-Sakaguchi oscillators [16]:

$$\dot{\theta}_i = \omega_i + \sum_{j=1}^n A_{ij} \sin(\theta_j - \theta_i + \varphi).$$

The phase shift φ makes the matrix formulation of the above dynamics to be incompatible with the ones we have introduced in equation (3) of the main text. We can instead write the above dynamics in matrix form by considering the graph \mathcal{G} as a directed graph (see also [R1:9] below), with \mathcal{E}_d being the set of directed edges. That is, we consider each undirected edge as two directed edges where each direction $(j, i) \neq (i, j)$ has the same weight $A_{ij} = A_{ji}$. Then, we can write the following equation:

$$\boldsymbol{\omega} + B_{\text{sink}} D_d(\bar{\mathbf{x}}_{\text{min}}, \varphi) [\boldsymbol{\delta}^T \boldsymbol{\delta}^T]^T = \omega_{\text{sync}} \mathbf{1}, \quad (3)$$

where $B_{\text{sink}} \in \mathbb{R}^{n \times |\mathcal{E}_d|}$ satisfies $B_{\text{sink}, k\ell} = -1$ if k is the sink of the interconnection ℓ and $B_{\text{source}, k\ell} = 0$ otherwise, $D_d \in \mathbb{R}^{|\mathcal{E}_d| \times |\mathcal{E}_d|}$ is the diagonal matrix of all $\sin(\bar{x}_{ij} + \varphi)$, and $\omega_{\text{sync}} \in \mathbb{R}$ is the synchronization frequency of phase-locked trajectories.

We are now ready to apply the equation (3) as a constraint in our numerical optimization routine. Without loss of generality, we set $\omega_{\text{sync}} = 0$, and apply the same method developed for the lossless case to the IEEE

Figure 4: **Error between the pre-fault functional pattern R and the functional pattern $R_{\text{recovered}}$ obtained through our procedure as a function of the phase shift $\varphi \in \mathbb{S}^1$.** The functional pattern $R_{\text{recovered}}$ is computed after a network correction due to a fault that occurs in the IEEE 39 test case between loads 13 and 14 (the same as in the main text). In the presence of a phase shift φ , the error between the desired functional pattern and the one associated with the network correction computed by our method remains small for small values of energy loss φ . The mean error is computed as $\langle \text{vec}(R) - \text{vec}(R_{\text{recovered}}) \rangle$, and the maximum error as $\|\text{vec}(R) - \text{vec}(R_{\text{recovered}})\|_{\infty}$, where $\text{vec}(\cdot)$ denotes the vectorization. Here, the parameter optimization does not explicitly account for the phase shift φ .

39 New England test case to compute the optimal correction after a fault occurs between loads 13 and 14 (the same as in the main text). For a loss $\varphi = 0.01$, the updated optimization is able to recover the pre-fault functional pattern with a mean error $\langle \text{vec}(R) - \text{vec}(R_{\text{recovered}}) \rangle = 0.072$, where $\text{vec}(\cdot)$ vectorizes the matrix (see also Fig. 5). This result improves upon the original method proposed for lossless networks by guaranteeing a satisfactory power flow recovery for losses φ that are one order of magnitude larger. Finally, we observe that for $\varphi > 0.01$ the fixed parameters of this specific system cause the phases to lose frequency synchronization even at operating conditions before the fault occurs.

Figure 5: **Nominal, post-fault, and recovered functional pattern in a network with lossy communications.** In all panels, the loss is fixed to $\varphi = 0.01$. **a** Functional pattern associated to nominal power flow in the IEEE 39 New England test case. **b** Functional pattern associated to a power flow disruption due to a fault that disconnects loads 13 and 14. **c** The recovered functional pattern after our procedure with updated constraint (equation (3)) is applied. The mean error between the pre-fault and the recovered functional patterns is $\langle \text{vec}(R) - \text{vec}(R_{\text{recovered}}) \rangle = 0.072$, where $\text{vec}(\cdot)$ denotes the vectorization of the patterns.

[R1: 9]

Minor remarks and comments:

The adjacency matrix is assumed to be symmetric which is another restriction of the formalism. Is it possible to lift this restriction?

Response: Good point. We find that, with a slight modification, our formalism can handle asymmetric adjacency matrices and, thus, directed networks. Such a modification consists of the rewriting of the main constraint in the optimization problems proposed in the main text.

By relaxing the assumption on symmetric adjacency matrices, the matrix form of an oscillator network with Kuramoto dynamics in equation (3) of the main text does not hold anymore and requires a rewriting. In what follows, we use the subscript “d” to indicate notation associated to *directed* graphs. Specifically, \mathcal{E}_d denote the oriented edge set, so that $D_d \in \mathbb{R}^{|\mathcal{E}_d| \times |\mathcal{E}_d|}$ and $\delta_d \in \mathbb{R}^{|\mathcal{E}_d|}$ denote the diagonal matrix of all $\sin(x_{ij})$ with $(j, i) \in \mathcal{E}_d$, and the vector of all the network weights $A(i, j) \neq 0$, respectively. Further, let us define $B_{\text{source}} \in \mathbb{R}^{n \times |\mathcal{E}_d|}$ the modified incidence matrix whose columns have nonzero entries only at the edges’ sources. That is, $B_{\text{source}, k\ell} = -1$ if k is the source of the interconnection ℓ , and $B_{\text{source}, k\ell} = 0$ otherwise. These definitions allow us to define the matrix form for a directed network of Kuramoto oscillators:

$$\dot{\boldsymbol{\theta}} = [\omega_1 \ \cdots \ \omega_n]^\top - B_{\text{source}} D_d(\mathbf{x}_{\min}) \delta_d. \quad (4)$$

The main change in the behavior of directed networks with respect to undirected ones is that the frequencies $\boldsymbol{\theta}$ of phase-locked trajectories do not typically converge to the average natural frequency (i.e., $\boldsymbol{\theta} \neq \omega_{\text{mean}} \mathbf{1}$). For phase-locked trajectories we have that $\boldsymbol{\theta} = \omega_{\text{sync}} \mathbf{1}$, where the constant $\omega_{\text{sync}} \in \mathbb{R}$ is not known *a priori*, and can only be estimated in the almost-fully synchronized regime $|\theta_i(t) - \theta_j(t)| \ll 1$ for all $i, j \in \mathcal{O}$ [17]. Yet, not knowing ω_{sync} as we do in the undirected case does not prevent the definition of a framework that can enforce a target functional pattern. In fact, by using equation (4) we can concurrently achieve the functional pattern associated with $\mathbf{x}_{\text{desired}}$ and assign a desired phase-locked frequency ω_{sync} . To do so, we utilize equation (4) above with $\boldsymbol{\theta} = \omega_{\text{sync}} \mathbf{1}$ as a constraint in the optimization problems proposed in the main text. This modification allows us to extend any of the control methods to achieve desired functional relationships to directed networks.

As an example of optimization of the coupling strengths, we solve

$$\min_{\boldsymbol{\alpha}} \quad \|\boldsymbol{\alpha}\|_1 \quad (5)$$

$$\text{subject to} \quad [\omega_1 \ \cdots \ \omega_n]^\top - B_{\text{source}} D_d(\mathbf{x}_{\min}) (\delta_d + \boldsymbol{\alpha}) = \omega_{\text{sync}} \mathbf{1}, \quad (5a)$$

for a network of $n = 7$ oscillators with adjacency matrix and natural frequencies as follows:

$$A_d = \begin{bmatrix} 0 & \mathbf{0} & 1 & 1 & 1 & 1 & 1 \\ 1 & 0 & 1 & 1 & 1 & 1 & 1 \\ 1 & 1 & 0 & 1 & 1 & 1 & 1 \\ 1 & 1 & 1 & 0 & 1 & 1 & 1 \\ 1 & 1 & 1 & 1 & 0 & 1 & 1 \\ 1 & 1 & 1 & 1 & 1 & 0 & 1 \\ 1 & 1 & 1 & 1 & 1 & 1 & 0 \end{bmatrix} \quad \text{and} \quad \bar{\boldsymbol{\omega}} = 0.1 \cdot \begin{bmatrix} 1 \\ 2 \\ 3 \\ 4 \\ 5 \\ 6 \\ 7 \end{bmatrix},$$

where the entry highlighted in red is the one causing the asymmetry in the network coupling. The target phase differences x_{1i} are $\mathbf{x}_{\text{desired}} = [\frac{\pi}{3} \ \frac{\pi}{4} \ \frac{\pi}{6} \ \frac{\pi}{8} \ \frac{\pi}{8} \ \frac{\pi}{6}]^\top$, and the desired phase locking frequency is $\omega_{\text{sync}} = 1$ rad/sec. The solution $\boldsymbol{\alpha}^*$ to problem (5) above yields the asymmetric adjacency matrix

$$A_d = \begin{bmatrix} 0 & 0 & -1.2238 & 1 & 1 & 1 & 1 \\ -3.7832 & 0 & 1 & 1 & 1 & 1 & 1 \\ -2.4384 & 1 & 0 & 1 & 1 & 1 & 1 \\ 0.3978 & 1.6022 & 1 & 0 & 1 & 1 & 1 \\ 1 & 0.3925 & 1 & 1 & 0 & 1 & 1 \\ 1 & 0.2282 & 1 & 1 & 1 & 0 & 1 \\ 0.6978 & 1.3022 & 1 & 1 & 1 & 1 & 0 \end{bmatrix}.$$

Fig. 6 illustrates that the phase trajectories associated with such a solution achieve the target functional pattern.

We conclude this discussion by remarking that, besides our optimization problems, the sufficient conditions (i.a), (i.b), and (i.c) in the main text for the existence of positive coupling strengths that realize a target pattern

Figure 6: **Desired functional pattern and phase-locked frequency in a directed network.** **a** Phase trajectory of the modified network after the solution α^* to the problem in (5) is applied to a network of $n = 7$ oscillators with target phase differences x_{1i} equal to $\mathbf{x}_{\text{desired}} = [\frac{\pi}{3} \ \frac{\pi}{4} \ \frac{\pi}{6} \ \frac{\pi}{8} \ \frac{\pi}{8} \ \frac{\pi}{6} \ \frac{\pi}{3}]^T$ and desired phase locking frequency is $\bar{k}_{\text{freq}} = 1$. The initial conditions \mathbf{x}_0 are chosen randomly and satisfy $\|\mathbf{x}_0 - \mathbf{x}_{\text{desired}}\| < 0.5$. **b** The phase difference trajectories achieve the target values in $\bar{\mathbf{x}}_{\text{min}}$. **c** The phase trajectories achieve the functional pattern associated with the target phase differences.

can also be adapted to the case of directed networks. To show this, we let $\boldsymbol{\omega}_d = [\omega_1 - \omega_{\text{sync}} \ \cdots \ \omega_n - \omega_{\text{sync}}]^T$ and $\bar{B}_{\text{source}} = B_{\text{source}} \text{sign}(D_d(\mathbf{x}))$. Then, a sufficient condition for the existence of positive network weights that achieve a desired functional pattern is the following one.

There exists $\delta \geq 0$ such that $B_{\text{source}} D_d(\mathbf{x}) \delta_d = \boldsymbol{\omega}_d$ if there exists a set \mathcal{S} satisfying:

- (iii.a) $D_{d_{ii}}(\mathbf{x}) D_{d_{jj}}(\mathbf{x}) B_{\text{source},i}^T B_{\text{source},j} \leq 0$ for all $i, j \in \mathcal{S}$ with $i \neq j$;
- (iii.b) $\boldsymbol{\omega}_d^T B_{\text{source},i} D_{d_{ii}}(\mathbf{x}) > 0$ for all $i \in \mathcal{S}$;
- (iii.c) $\boldsymbol{\omega}_d \in \text{Im}(B_{\text{source},\mathcal{S}})$.

Note that, since ω_{sync} is not known *a priori* in most cases, these conditions are hard to check.

We have modified the main text to point out that our control methods can be extended to directed networks in the *Introduction* and at the end of section *Optimal interventions for desired functional patterns*. We have also added the above detailed discussion to the Supplementary Information.

[R1: 10] *How are the parameters $c1$ and $c2$ chosen in the power grid example?*

Response: Scaling parameters are set to 1 in the cost functional for this example. We have clarified this in the main text.

[R1: 11] *I think (i.b) implies (i.c). Please clarify.*

Response: Let us recall below the conditions for the existence of positive network weights that satisfy $BD(\mathbf{x}_{\text{desired}}) \delta^* = \boldsymbol{\omega}$:

There exists $\delta \geq 0$ such that $BD(\mathbf{x}) \delta = \boldsymbol{\omega}$ if there exists a set \mathcal{S} satisfying:

- (i.a) $D_{ii}(\mathbf{x}) D_{jj}(\mathbf{x}) B_{:,i}^T B_{:,j} \leq 0$ for all $i, j \in \mathcal{S}$ with $i \neq j$ and $D_{ii}, D_{jj} \neq 0$;
- (i.b) $\boldsymbol{\omega}^T B_{:,i} D_{ii}(\mathbf{x}) > 0$ for all $i \in \mathcal{S}$;
- (i.c) $\boldsymbol{\omega} \in \text{Im}(B_{:, \mathcal{S}})$.

Notice that condition (i.b) requires $\boldsymbol{\omega}^\top \bar{B}_{:,S} > 0$, and condition (i.c) requires that $\boldsymbol{\omega} \in \text{Im}(B_{:,S}) = \text{Im}(\bar{B}_{:,S})$. Since $\boldsymbol{\omega}$ may have positive projections on the columns of $\bar{B}_{:,S}$ but not belong to its image, (i.b) does not imply (i.c).

As an example, consider $\bar{B}_{:,S}$ as a single vector in \mathbb{R}^n . Even if (i.b) holds and $\boldsymbol{\omega}$ has a positive projection on $\bar{B}_{:,S}$, $\boldsymbol{\omega}$ should also be parallel to $\bar{B}_{:,S}$ for (i.c) to hold.

[R1: 12] *Please add some details, how the last equation of page 6 is derived. It took me some minutes to get what is meant by the expressions in the equation.*

Response: Thank you for the suggestion. We have improved the reasoning and explanations for the existence of a strictly positive solution in networks that satisfy conditions (ii.a) and (ii.b). We have expanded the main text to clarify our result. The main text now reads:

“By rewriting the pattern assignment problem as in equation (6), the following vector of interconnection weights solves such equation (see Methods):

$$\boldsymbol{\delta}_{\mathcal{H}} = (B_{:, \mathcal{H}} D_{\mathcal{H}, \mathcal{H}}(\mathbf{x}))^\dagger \left(\boldsymbol{\omega} - B_{:, \tilde{\mathcal{H}}} D_{\tilde{\mathcal{H}}, \tilde{\mathcal{H}}}(\mathbf{x}) \boldsymbol{\delta}_{\tilde{\mathcal{H}}} \right).$$

Because $\bar{B}_{:, \mathcal{H}} = B_{:, \mathcal{H}} D_{\mathcal{H}, \mathcal{H}}$ defines an Hamiltonian path and because of (ii.b), the vector $(B_{:, \mathcal{H}} D_{\mathcal{H}, \mathcal{H}}(\mathbf{x}))^\dagger \boldsymbol{\omega}$ contains only strictly positive entries. Thus, for any sufficiently small and positive vector $\boldsymbol{\delta}_{\tilde{\mathcal{H}}}$, the weights $\boldsymbol{\delta}_{\mathcal{H}}$ are also strictly positive, ultimately proving the existence of a strictly positive solution to the pattern assignment problem (see Methods).”

The Methods section containing the technical details reads:

“Rewrite the pattern assignment problem $BD(\mathbf{x})\boldsymbol{\delta} = \boldsymbol{\omega}$ as

$$BD(\mathbf{x})\boldsymbol{\delta} = B_{:, \mathcal{H}} D_{\mathcal{H}, \mathcal{H}}(\mathbf{x}) \boldsymbol{\delta}_{\mathcal{H}} + B_{:, \tilde{\mathcal{H}}} D_{\tilde{\mathcal{H}}, \tilde{\mathcal{H}}}(\mathbf{x}) \boldsymbol{\delta}_{\tilde{\mathcal{H}}} = \boldsymbol{\omega},$$

where the subscripts \mathcal{H} and $\tilde{\mathcal{H}}$ denote the entries corresponding to the Hamiltonian path in conditions (ii.a)-(ii.b) and the remaining ones, respectively. Since $\text{Im}(\bar{B}_{:, \mathcal{H}}) = \text{Im}(B_{:, \mathcal{H}} D_{\mathcal{H}, \mathcal{H}}(\mathbf{x})) = \text{span}(\mathbf{1})^\perp$, $\text{Im}(B_{:, \tilde{\mathcal{H}}} D_{\tilde{\mathcal{H}}, \tilde{\mathcal{H}}}(\mathbf{x})) \subseteq \text{span}(\mathbf{1})^\perp$, and $\boldsymbol{\omega} \in \text{Im}(\mathbf{1})^\perp$, for any vector $\boldsymbol{\delta}_{\tilde{\mathcal{H}}}$, the following set of weights solves the above equation:

$$\begin{aligned} \boldsymbol{\delta}_{\mathcal{H}} &= (B_{:, \mathcal{H}} D_{\mathcal{H}, \mathcal{H}}(\mathbf{x}))^\dagger \left(\boldsymbol{\omega} - B_{:, \tilde{\mathcal{H}}} D_{\tilde{\mathcal{H}}, \tilde{\mathcal{H}}}(\mathbf{x}) \boldsymbol{\delta}_{\tilde{\mathcal{H}}} \right) \\ &= (D_{\mathcal{H}, \mathcal{H}}(\mathbf{x}) B_{:, \mathcal{H}}^\top B_{:, \mathcal{H}} D_{\mathcal{H}, \mathcal{H}}(\mathbf{x}))^{-1} D_{\mathcal{H}, \mathcal{H}}(\mathbf{x}) B_{:, \mathcal{H}}^\top \left(\boldsymbol{\omega} - B_{:, \tilde{\mathcal{H}}} D_{\tilde{\mathcal{H}}, \tilde{\mathcal{H}}}(\mathbf{x}) \boldsymbol{\delta}_{\tilde{\mathcal{H}}} \right) \end{aligned}$$

Because the matrix $D_{\mathcal{H}, \mathcal{H}}(\mathbf{x}) B_{:, \mathcal{H}}^\top B_{:, \mathcal{H}} D_{\mathcal{H}, \mathcal{H}}(\mathbf{x})$ is an M-matrix, its inverse has nonnegative entries. Further, by condition (ii.b), $D_{\mathcal{H}, \mathcal{H}}(\mathbf{x}) B_{:, \mathcal{H}}^\top \boldsymbol{\omega}$ is strictly positive. Then, the vector

$$(D_{\mathcal{H}, \mathcal{H}}(\mathbf{x}) B_{:, \mathcal{H}}^\top B_{:, \mathcal{H}} D_{\mathcal{H}, \mathcal{H}}(\mathbf{x}))^{-1} D_{\mathcal{H}, \mathcal{H}}(\mathbf{x}) B_{:, \mathcal{H}}^\top \boldsymbol{\omega}$$

is also strictly positive, and so is the solution vector $\boldsymbol{\delta}_{\mathcal{H}}$ for any sufficiently small and positive vector $\boldsymbol{\delta}_{\tilde{\mathcal{H}}}$.”

[R1: 13] *Is there a bound on how many patterns can be prescribed in the "multi-pattern" optimization problem for a given system with n oscillators?*

Response: Thank you for another important question. In general, there does not exist an upper bound to the number of patterns that can be prescribed. We elaborate in the main text that the relationship between the network structure and the kernel of its incidence matrix enables the characterization of which (and how many) compatible patterns coexist. The main text now reads:

“A single choice of the interconnections weights can allow for multiple desired functional patterns, as long as they are compatible with the network dynamics in equation (2). In this section, we provide a characterization

of compatible functional patterns in a given network, and derive conditions for the existence of a set of interconnection weights that achieve multiple desired functional patterns. Being able to concurrently assign multiple functional patterns is crucial, for instance, to the investigation and design of memory systems [?], where different patterns of activity correspond to distinct memories. Furthermore, our results complement previous work on the search of equilibria in oscillator networks [3].

To find a set of functional patterns that exists concurrently in a given network with fixed interconnection weights $\boldsymbol{\delta}$, we exploit the algebraic core of equation (4) and show that the kernel of the incidence matrix B uniquely determines the equilibria of the network. In fact, for a given network (i.e., $\boldsymbol{\delta}$ with nonzero components) all compatible equilibria $\boldsymbol{x}^{(i)}$, $i \in \{1, \dots, \ell\}$, must satisfy

$$D(\boldsymbol{x}^{(i)})\boldsymbol{\delta} = B^\dagger\boldsymbol{\omega} + \ker(B). \quad (6)$$

From equation (6), we can see that the sine vector of all compatible equilibria must belong to a specific affine subspace of $\mathbb{R}^{|\mathcal{E}|}$:

$$\sin(\boldsymbol{x}^{(i)}) \in \text{diag}(\boldsymbol{\delta})^{-1} (B^\dagger\boldsymbol{\omega} + \ker(B)). \quad (7)$$

Rewriting equation (4) in the above form connects the existence of distinct functional patterns with $\ker(B)$, the latter featuring a number of well known properties. For instance, it holds that $\dim(\ker(B)) = |\mathcal{E}| - n + c$, where c is the number of connected components in a network, and that $\ker(B)$ coincides with the subspace spanned by the signed path vectors of all undirected cycles in the network [36]. Notice also that, after a suitable reordering of the phase differences in \boldsymbol{x} , we can write $\sin(\boldsymbol{x}) = [\sin(\boldsymbol{x}_{\text{desired}}^\top) \ \sin(\boldsymbol{x}_{\text{dep}}^\top)]^\top$, where $\boldsymbol{x}_{\text{dep}}$ denotes the phase differences dependent on $n - 1$ desired phase differences $\boldsymbol{x}_{\text{desired}}$. Thus, all the $\boldsymbol{x}_{\text{desired}}$ for which $\sin(\boldsymbol{x}_{\text{dep}})$ intersects the affine space described by $\text{diag}(\boldsymbol{\delta})^{-1} (B^\dagger\boldsymbol{\omega} + \ker(B))$ identify compatible functional patterns.

To showcase how the intimate relationship between the network structure and the kernel of its incidence matrix enables the characterization of which (and how many) compatible patterns coexist, we consider three essential network topologies: trees, cycles, and complete graphs. For the sake of simplicity, we let $\boldsymbol{\delta} = \mathbf{1}$ and $\boldsymbol{\omega} = \mathbf{0}$, so that equation (7) holds whenever $\sin(\boldsymbol{x}^{(i)}) \in \ker(B)$. In networks with tree topologies it holds $\ker(B) = \mathbf{0}$, and $\sin(\boldsymbol{x}^{(i)}) = \mathbf{0}$ is satisfied by 2^{n-1} patterns of the form $x_{jk}^{(i)} = 0, \pi$, for all $(j, k) \in \mathcal{E}$. Consider now cycle networks, where $\ker(B) = \text{span}(\mathbf{1})$. For any cycle of $n \geq 3$ oscillators, two families of patterns are straightforward to derive. First, there are 2^{n-1} patterns of the form $\boldsymbol{x}_{k,k+1}^{(i)} = 0, \pi$, with $k = 1, \dots, n - 1$, and $x_{n1}^{(i)} = -\sum_{k=1}^{n-1} x_{k,k+1}^{(i)}$. Second, there are $n - 1$ splay states [4], where the oscillators' phases evenly span the unitary circle, with $x_{j,k}^{(i)} = \frac{2\pi m}{n}$, $m = 1, \dots, n - 1$, $(j, k) \in \mathcal{E}$. Fig. 7 illustrates the compatible functional patterns satisfying equation (7) in a positive network of three fully synchronizing oscillators. In general, however, cycle networks of identical oscillators admit infinite coexisting patterns. For instance, Fig. 8 shows how we can parameterize infinite equilibria with a scalar $\gamma \in \mathbb{S}^1$ in a cycle of $n = 4$ oscillators. Finally, as complete graphs are equivalent to a composition of cycles, they also admit infinite compatible patterns that can be parameterized akin to what occurs in a simple cycle (see Supplementary Figure 2)."

[R1: 14] In power grids the natural frequency is given by $\omega = p/D$ where D is the damping constant.

Response: Thank you for pointing this out. We have clarified this imprecision in the manuscript.

[R1: 15] Suppl. Mat.: In Thm 1, it should read $[\gamma, \phi_2, \dots, \phi_{n-1}]$. In the proof, do you distinguish between n odd and n even?

Response: All the phase differences in $\boldsymbol{x}_{\text{cycle}}$ represent phase differences between connected oscillators. Thus, the equilibrium $\boldsymbol{x}_{\text{cycle}}$ contains n phase differences, not $n - 1$ (differently from the main text, where $\boldsymbol{x}_{\text{desired}}$ contain $n - 1$ phase differences). We choose not to use $\boldsymbol{x}_{\text{desired}}$ in this theorem to highlight the fact that all phase differences x_{ij} satisfying $|x_{ij}| < \frac{\pi}{2}$ are pairwise identical. To avoid confusion, the text before Theorem 1 now explicitly says:

Figure 7: The intersection of an affine space with $\sin(\mathbf{x}_{\text{dep}})$ determines the compatible functional patterns of 3 identical oscillators. Consider a fully connected network of $n = 3$ identical oscillators with zero natural frequency and $\delta = \mathbf{1}$. It is well known that $\mathbf{x}^{(1)} = [0 \ 0]^\top$, $\mathbf{x}^{(2)} = [\pi \ 0]^\top$, $\mathbf{x}^{(3)} = [0 \ \pi]^\top$, $\mathbf{x}^{(4)} = [\pi \ \pi]^\top$ are phase difference equilibria. Furthermore, because $\sin(\theta) = \sin(\pi - \theta)$, this figure illustrates $\sin(x_{13})$ as a function of x_{12} and x_{23} in four different panels: $\sin(x_{12} + x_{23})$ (top left), $\sin(\pi - x_{12} + x_{23})$ (top right), $\sin(x_{12} + \pi - x_{23})$ (bottom left), and $\sin(-x_{12} - x_{23})$ (bottom right). The fourth panel reveals that the two functional patterns compatible with $\mathbf{x}^{(j)}$, $j \in \{1, \dots, 4\}$, correspond to $\mathbf{x}^{(5)} = [2\pi/3 \ 2\pi/3]^\top$ and $\mathbf{x}^{(6)} = [-2\pi/3 \ -2\pi/3]^\top$ (in red).

“Consider a cycle network of $n > 4$ oscillators with positive weights, and denote with $\mathbf{x}_{\text{cycle}}$ a vector of the n phase differences between connected oscillators.”

Finally, the proof does not distinguish between n odd and n even. We have removed the erroneous sentence that caused confusion.

[R1: 16] *Suppl. Mat.: There are two zeros in front of the dot in Eq.(8), please drop one zero.*

Response: Fixed.

Figure 8: **A homogeneous cycle network admits infinite compatible functional patterns.** Since $\ker(B) = \text{span } \mathbf{1}$, the cycle network admits infinite compatible equilibria, which can be parameterized by $\gamma \in \mathbb{S}^1$ as $\mathbf{x}^{(i)}(\gamma) = [\frac{\pi}{2} - \gamma, \frac{\pi}{2} + \gamma, \frac{\pi}{2} - \gamma, \frac{\pi}{2} + \gamma]^T$. Any arbitrarily small variation of γ yields $\sin(\mathbf{x}^{(i)}(\gamma)) \in \ker(B)$. The right panel illustrates the patterns associated with $\mathbf{x}^{(i)}(\gamma)$, $i = 1, \dots, 5$ for increments of γ of 0.2 radians.

Response to Reviewer 2

We thank the reviewer for their constructive comments. Note that all the changes following this reviewer’s comments are highlighted in blue in the revised manuscript.

[R2: 1] *This paper studies the control of functional patterns in oscillatory networks. The work is based on considering phase models, and the main contribution is to develop principled method to enforce desired functional patterns by optimal tuning of oscillator parameters. Overall, the paper is well structured, and the problem of functional control is relevant and important to the field of oscillatory networks.*

Response: We thank this reviewer for providing insightful comments that have helped us revise and considerably improve our work.

For this reviewer’s convenience, we list below the main additions and changes to the manuscript that regard this reviewer’s suggestions.

The major additions to the manuscript are:

- (a) We have added a section in the Supplementary Information to motivate the choice of the Kuramoto model to approximate fMRI time series;
- (b) We have added a technical discussion and examples in the Supplementary Information to motivate the choice of our correlation metric in place of the classical Pearson correlation coefficient.

We have also rewritten the part “*Analytical Results*”. Besides addressing this reviewer’s comments, this part now contains novel results (particularly with respect to the coexistence of multiple functional patterns), a simplified notation, a clearer exposition, and more examples. With respect to the reviewers’ comments, the major changes we made in the revised version are as follows.

- (a) We have enhanced the text in the section “*Feasible functional patterns in positive networks*” that explains why conditions (ii.a) and (ii.b) are sufficient for the existence of a strictly positive solution. We have also added a section in the Methods containing the technical details used to demonstrate our claim;
- (b) We have expanded the derivation of the equations concerning the feasibility of desired functional patterns in strictly positive networks possessing Hamiltonian paths;

- (c) We have extended and enhanced our treatment of the stability of functional patterns. The main manuscript contains now a more comprehensive description and an application of our heuristic method to promote stability of functional patterns with negative correlations in positive networks.

[R2: 2]

However, the overall contribution of the work is very limited. The control of functional patterns, e.g., phase-locking, synchronization, or cluster patterns, in oscillatory networks has been extensively studied. In particular, existing works on optimal control and optimization for phase assignment using phase-models can be widely found in the literature.

Response: We agree with the reviewer that there exist many established methods to achieve phase locking (i.e., frequency synchronization), full synchronization, and cluster synchronization. However, these methods achieve desired macroscopic observables of network synchronization (e.g., a certain global synchronization degree). In contrast, our method allows to prescribe specific pairwise relations, which is the most general approach to synchronization problems. Being able to control pairwise functional relationships between the oscillators solves problems such as phase-locking, full, and cluster synchronization, but the converse is not true. In fact, even in the general case of cluster synchronization (where $m \geq 1$ groups of cohesive oscillators coexist in the network), one can only achieve a desired synchronization level *within* the clusters [19], but not *across* clusters, which instead is possible with our approach. Further, most control methods apply exogenous signals, while our control scheme tunes the network parameters. Thus, we still believe that our paper addresses an important and challenging problem with a novel approach.

The work that is closest to ours is [20], where the authors seek for interconnection weights and natural frequencies of Kuramoto oscillators to achieve a specified level of phase cohesiveness in a network of heterogeneous Kuramoto oscillators. Since the only goal of the aforementioned work is to bound the phase differences, we believe that our work significantly departs from [20]. Specifically, we enable the prescription of pairwise phase differences, and investigate the stability properties of all desired functional patterns. Further, an early work [21] links the network structure (in terms of power law exponent and degree correlation coefficient) to the network-wide level of synchronization of the network, and proposes a method to optimize the oscillators' degree correlation. However, similarly to [20], the control strategy proposed in [21] falls short of enabling the microscopic (i.e., pairwise) functional relationships that instead our work allows to specify.

We have extended the section *Contributions and future directions* to emphasize our contributions with respect to existing literature as follows:

“An important consideration that highlights the general contributions of the present study is that being able to specify pairwise functional relationships between the oscillators also solves problems such as phase-locking, full, and cluster synchronization; yet, the converse is not true. In fact, even in the general setting of cluster synchronization [22] – where distinct groups of oscillators behave cohesively – one can only achieve a desired synchronization level *within* the same cluster, but not *across* clusters, which instead is possible with our approach. Specifically, oscillators belonging to the same cluster are forced to synchronize, thus implying that the associated diagonal blocks of the functional pattern R display values close to 1. Seminal work in Ref. [23] developed a nonlinear feedback control to change the coupling functions in equation (2) to engineer clusters of synchronized oscillators, whereas the authors of Ref. [24] propose the formation of clusters through selective addition of edges to the network. More recently, the control of partially synchronized states with applications to brain networks is studied in Ref. [19] by means of structural interventions, and in Ref. [25] via exogenous stimulation.”

The comparison with the work in Ref. [20] now appears at the end of the introduction as:

“While synchronization phenomena in oscillator networks have been studied extensively (e.g., see [26, 27, 21, 28, 29]), the development of control methods to impose desired synchronous behaviors has only recently attracted the attention of the research community [30, 20, 31, 19]. Perhaps the work that is closest to our approach is Ref. [20], where the authors tailor interconnection weights and natural frequencies to achieve a specified level of phase cohesiveness in a network of Kuramoto oscillators. Our work improves considerably upon this latter study, whose goal is limited to prescribing an upper bound to the phase differences, by enabling the prescription of pairwise phase differences and by investigating the stability properties of different functional

patterns. Taken together, existing results highlight the importance of controlling distinct configurations of synchrony, but remain mainly focused on the control of “macroscopic” observables of synchrony (e.g., the average synchronization level of all the oscillators). In contrast, our control method prescribes desired pairwise levels of correlation across all of the oscillators, thus enabling a “microscopic” description of functional interactions.”

We have done our best to review relevant literature. We would also be happy to review any specific suggestions.

[R2: 3] *In addition, the proposed formulations and techniques for functional/pattern control by tuning network weights and natural frequencies of oscillators are quite standard in control systems. One unique point addressed in this work is the consideration of positive couplings, which is practically relevant, however, it can be easily accommodated as a constraint in the formulated optimization problem as done in the paper.*

Response: In regards to the novelty of our method to prescribe synchronization patterns in oscillator networks, we remark that our results significantly depart from more classical synchronization control techniques such as pinning [32], feedback for formation control [33], and stabilization of phase differences equilibria [34]. Another major difference from classical control schemes is that we do not require the oscillators’ phases to be measurable, as our procedure relies on some system’s parameters being the control knobs to achieve the desired functional pattern. To the best of our knowledge, methods to control the structure of a nonlinear network are much less investigated, if at all present.

In addition, here we do not stop at simply formulating the long-standing problem of synchronization control in a general fashion that cleverly exploits convex constraints, but we also explore the structure of the solutions. Specifically, we provide graph-theoretic and algebraic insights that link the existence and the stability of functional patterns to the network structure and parameters. As highlighted by our analytical insights and the applications to complex networks, our study goes well beyond the simple formulation of constraints in an optimization problem.

[R2: 4] *The paper is mostly technically sound, and the content is much more suitable to a specialized domain journal, e.g., in control systems or nonlinear science. The proposed analytical results, including the optimal control and convex optimization formulations and approaches, as well as the stability analysis are within the domain of classical control systems.*

Response: We believe that the paper suits the scope and audience of Nature Communications for the following reasons:

- The generality of the problem, which is often covered in Nature Communications (see, e.g., [35, 22, 12]);
- The generality of our applications, which is appreciated more by a broader audience than the one in strictly technical journals;
- We present insightful and general analysis and synthesis techniques, which are of interest and potentially applicable to other problems. While the problem of controlling synchronization metrics may be a classical one, our paper provides novel yet mature results. Specifically, we provide techniques that allow us to *understand* the link between structure and function in oscillator networks by characterizing the existence and the stability of the solutions through algebraic and graph-theoretic tools. This is in stark contrast to previous literature where, although having an objective coarser than ours, the interpretation of the results is hindered by the chosen methods (see., e.g., [20, 19, 34]).

[R2: 5] *Also, the presented applications to complex networks are limited to Kuramoto oscillators.*

Response: The applications in the paper use Kuramoto oscillators because our analysis is performed for this model. We emphasize that Kuramoto oscillators have been used to model a wide range of phenomena

in numerous scientific domains. Examples include power systems, Josephson junctions, neuronal networks, pacemaker cells in the heart, circadian cells, coupled analog clocks, and flocking, schooling, and vehicle coordination (see, e.g., the surveys [8, 4] and references therein). Thus, our model covers a large class of systems.

Further, our results perform well even with more general models. To emphasize this point, we have included new analyses and simulations on higher-order and more realistic power systems models in the revised version of the paper (see also [R1: 8] above). More in detail, we show that, in the case of power flow recovery, first-order Kuramoto dynamics are still an effective approximation of third-order (or q -axis) and leaky power network models. Clearly, if an oscillatory system has dynamics that are quantitatively and qualitatively different from the ones of Kuramoto oscillators, our methods may not be applicable.

[R2: 6]

Here are some specific comments:

As the paper has a strong technical flavor, there is no rigorous proof of the conditions that guarantee the existence of the solution of the linear inverse problem. Especially, on Page 6, it is incorrect to argue the existence of a positive solution by saying merely that an arbitrary small solution can be found.

Response: We apologize for the lack of details to support our claims in the initial submission of the manuscript. We have improved the reasoning and explanations for the existence of a positive and strictly positive solution in networks that satisfy conditions (ii.a) and (ii.b). Specifically, we have expanded the main text, which now reads:

“By rewriting the pattern assignment problem as in equation (6), the following vector of interconnection weights solves such equation (see Methods):

$$\boldsymbol{\delta}_{\mathcal{H}} = (B_{:, \mathcal{H}} D_{\mathcal{H}, \mathcal{H}}(\mathbf{x}))^{\dagger} \left(\boldsymbol{\omega} - B_{:, \tilde{\mathcal{H}}} D_{\tilde{\mathcal{H}}, \tilde{\mathcal{H}}}(\mathbf{x}) \boldsymbol{\delta}_{\tilde{\mathcal{H}}} \right).$$

Because $\bar{B}_{:, \mathcal{H}} = B_{:, \mathcal{H}} D_{\mathcal{H}, \mathcal{H}}$ defines an Hamiltonian path and because of (ii.b), the vector $(B_{:, \mathcal{H}} D_{\mathcal{H}, \mathcal{H}}(\mathbf{x}))^{\dagger} \boldsymbol{\omega}$ contains only strictly positive entries. Thus, for any sufficiently small and positive vector $\boldsymbol{\delta}_{\tilde{\mathcal{H}}}$, the weights $\boldsymbol{\delta}_{\mathcal{H}}$ are also strictly positive, ultimately proving the existence of a strictly positive solution to the pattern assignment problem (see Methods).”

The Methods section containing the technical details reads:

“Rewrite the pattern assignment problem $BD(\mathbf{x})\boldsymbol{\delta} = \boldsymbol{\omega}$ as

$$BD(\mathbf{x})\boldsymbol{\delta} = B_{:, \mathcal{H}} D_{\mathcal{H}, \mathcal{H}}(\mathbf{x})\boldsymbol{\delta}_{\mathcal{H}} + B_{:, \tilde{\mathcal{H}}} D_{\tilde{\mathcal{H}}, \tilde{\mathcal{H}}}(\mathbf{x})\boldsymbol{\delta}_{\tilde{\mathcal{H}}} = \boldsymbol{\omega},$$

where the subscripts \mathcal{H} and $\tilde{\mathcal{H}}$ denote the entries corresponding to the Hamiltonian path in conditions (ii.a)-(ii.b) and the remaining ones, respectively. Since $\text{Im}(\bar{B}_{:, \mathcal{H}}) = \text{Im}(B_{:, \mathcal{H}} D_{:, \mathcal{H}}(\mathbf{x})) = \text{span}(\mathbf{1})^{\perp}$, $\text{Im}(B_{:, \tilde{\mathcal{H}}} D_{:, \tilde{\mathcal{H}}}(\mathbf{x})) \subseteq \text{span}(\mathbf{1})^{\perp}$, and $\boldsymbol{\omega} \in \text{span}(\mathbf{1})^{\perp}$, for any vector $\boldsymbol{\delta}_{\tilde{\mathcal{H}}}$, the following set of weights solves the above equation:

$$\begin{aligned} \boldsymbol{\delta}_{\mathcal{H}} &= (B_{:, \mathcal{H}} D_{\mathcal{H}, \mathcal{H}}(\mathbf{x}))^{\dagger} \left(\boldsymbol{\omega} - B_{:, \tilde{\mathcal{H}}} D_{\tilde{\mathcal{H}}, \tilde{\mathcal{H}}}(\mathbf{x}) \boldsymbol{\delta}_{\tilde{\mathcal{H}}} \right) \\ &= (D_{\mathcal{H}, \mathcal{H}}(\mathbf{x}) B_{:, \mathcal{H}}^{\top} B_{:, \mathcal{H}} D_{\mathcal{H}, \mathcal{H}}(\mathbf{x}))^{-1} D_{\mathcal{H}, \mathcal{H}}(\mathbf{x}) B_{:, \mathcal{H}}^{\top} \left(\boldsymbol{\omega} - B_{:, \tilde{\mathcal{H}}} D_{\tilde{\mathcal{H}}, \tilde{\mathcal{H}}}(\mathbf{x}) \boldsymbol{\delta}_{\tilde{\mathcal{H}}} \right) \end{aligned}$$

Because the matrix $D_{\mathcal{H}, \mathcal{H}}(\mathbf{x}) B_{:, \mathcal{H}}^{\top} B_{:, \mathcal{H}} D_{\mathcal{H}, \mathcal{H}}(\mathbf{x})$ is an M-matrix, its inverse has nonnegative entries. Further, by condition (ii.b), $D_{\mathcal{H}, \mathcal{H}}(\mathbf{x}) B_{:, \mathcal{H}}^{\top} \boldsymbol{\omega}$ is strictly positive. Then, the vector

$$(D_{\mathcal{H}, \mathcal{H}}(\mathbf{x}) B_{:, \mathcal{H}}^{\top} B_{:, \mathcal{H}} D_{\mathcal{H}, \mathcal{H}}(\mathbf{x}))^{-1} D_{\mathcal{H}, \mathcal{H}}(\mathbf{x}) B_{:, \mathcal{H}}^{\top} \boldsymbol{\omega}$$

is also strictly positive, and so is the solution vector $\boldsymbol{\delta}_{\mathcal{H}}$ for any sufficiently small and positive vector $\boldsymbol{\delta}_{\tilde{\mathcal{H}}}$.”

[R2: 7]

On Page 7, in the optimization problem in (7), which is an extension of the one in (6) on the same page, why is the positivity constraint dropped?

Response: We have improved the exposition of the entire section, and the problem which was in (7) is now in (11) in the revised text. The optimization problem in (11) represents the most general scenario, where *any* parameter of the network system can be modified and there are no constraints on the sign of the coupling strengths between the oscillators. To clarify this difference from the optimization in (9), we have modified the main text as follows:

“Note that the minimization problems (11) and (13) do not allow us to tune the oscillators’ natural frequencies, and are constrained to networks with positive weights. When any parameter of the network is unconstrained and can be adjusted to enforce a desired functional pattern, the network optimization problem can be generalized as

$$\begin{aligned} \min_{\alpha, \beta} \quad & \|[\alpha^\top \ \beta^\top]\|_2 & (8) \\ \text{subject to} \quad & BD(\mathbf{x})(\delta + \alpha) = [\omega_1 \ \cdots \ \omega_n]^\top + \beta, & (8a) \end{aligned}$$

where β denotes the correction to the natural frequencies. The minimization problem (8) always admits a solution because β can be chosen to satisfy the constraint (8a) for any choice of the network parameters $\delta + \alpha$. Further, the (unique) solution to the minimization problem (8) can also be computed in closed form:

$$\begin{bmatrix} \alpha^* \\ \beta^* \end{bmatrix} = [BD(\mathbf{x}) \quad -I_n]^\dagger ([\omega_1 \ \cdots \ \omega_n]^\top - BD(\mathbf{x})\delta)$$

where I_n denotes the $n \times n$ identity matrix.”

[R2: 8] *As shown in the example in Fig. 1, the corrected network has fewer edges than the original one, and hence during the phase assigning procedure, the rank of the matrix B in (7) may be decreased, does this impact on the feasibility of the proposed method?*

Response: The feasibility of our optimization problems is only affected by the initial parameters: topology and natural frequencies values. If a solution exists, it may disconnect some edges, and for more sparse networks the rank of the incidence matrix B may decrease to less than $n - 1$.³ Yet, whenever a solution exists (in terms of network parameters), the desired functional pattern is achieved, regardless of the sparsity of the adjusted network.

We would like to point out that the above discussion only concerns the invariance of a desired pattern. However, should one also be concerned with the stability of a desired pattern, the lack of connectivity could hinder stability, as we show in the section “Stability of functional patterns”. The loss of connectivity can be prevented by imposing constraints such strictly positive weights (i.e. $\delta + \alpha > 0$).

[R2: 9] *In the proposed method, it is crucial to find a Hamiltonian path that represents the solution. Does every Hamiltonian path generate the same result? If not, then how to apply the proposed method, e.g., in the case that the network is sparsely connected such that the desired Hamiltonian path is not in the network.*

Response: The existence of multiple Hamiltonian paths whose incidence matrices also have positive projections of the natural frequencies (i.e., they satisfy also (ii.b)) may give rise to different solutions to the optimization problem. However, they all yield the same functional pattern. This is because there may be multiple minimizers to the optimization problem, so that multiple networks can give rise to the same functional pattern – which is the ultimate goal of our procedure. Thus, *any* Hamiltonian path for which also (ii.b) is satisfied yields a network that is a solution to the optimization.

We remark that the existence of a Hamiltonian path whose incidence matrix displays positive projections of the natural frequency vector is only a sufficient condition for the feasibility of the optimization problem, but provides an interesting graph-theoretic insight into how the network structure dictates the problem feasibility.

³The directed incidence matrix of a connected network satisfies $\text{rank}(B) = n - 1$ [36].

We have added the following explanation to the main text after introducing the first optimization problem:

“Notice that, if a network contains multiple Hamiltonian paths whose incidence matrices also satisfy condition (ii.b), the minimization problem (6) may have multiple solutions α^* . However, such solutions yield the same desired functional pattern, showing that different network structures can give rise to the same functional pattern.”

[R2: 10] *In the application to brain networks, the authors claim that Kuramoto model gives a good approximate to the rhythmic activity of brain networks. However, there is no verification that the fMRI data fit the model well so that the results obtained by the proposed method is convincing.*

Response: One of the main outstanding problems in neuroscience is to find the best models to represent neural activity and brain functions. Here, we rely on previous extensive literature that uses Kuramoto phase oscillators to model fMRI data [37, 38, 39, 40, 41, 42, 19].

We have added a section in the Supplementary Information (*Coupled Kuramoto oscillators to approximate fMRI data*) to motivate our choice of the Kuramoto model:

“The interaction between static large-scale structural architecture of the human brain and local oscillations of neural communities is a key factor in the functional connectivity patterns that are empirically observed through functional magnetic resonance imaging (fMRI) when the brain is in a resting-state condition [43]. In the last two decades, extensive literature has resorted to Kuramoto phase oscillators to model fMRI data [37, 38, 39, 40, 41, 42, 19]. Many works, such as Ref. [38] and Ref. [40], focus on the analysis of the oscillatory behaviors of neural populations that lead the emergence of functionally connected networks by modeling fMRI data as the output of networks of Kuramoto oscillators. The main working assumption is that at each node of the structural brain network exists a community of excitatory and inhibitory neurons whose dynamical state is in a regime of self-sustained oscillations. From a modeling standpoint, this assumption is equivalent to employing a network of weakly coupled Wilson-Cowan oscillators [44, 45], or to a supercritical Andronov-Hopf bifurcation, such as the Stuart-Landau model in oscillatory regime [46]. In this setting, the neurons’ firing rates describe a closed periodic trajectory in phase space; that is, the firing rates delineate a limit cycle. Thus, the dynamics can be approximated by a single variable, which is the angle (or *phase*) on this cycle. This regime is then modeled by a network of coupled heterogeneous Kuramoto oscillators that are connected to each other according to the architecture of the human brain.”

[R2: 11] *The authors mention that the concept of functional pattern proposed in this work is a good replacement of the classical Pearson correlation. What are the benefits of using functional pattern instead of the Pearson correlation? This should be justified.*

Response: Good point. In this work, we provide methods and principles to guarantee the emergence of phase-locked trajectories that are then associated to functional patterns. The latter are defined utilizing $\rho_{ij} = \langle \cos(\theta_j - \theta_i) \rangle_t$. This metric is also utilized in [47] to quantify the similarity between BOLD signals in functional MRI recordings, and in [48], to quantify the synchrony in hierarchical networks.

Recall that the classical (sample) Pearson correlation, which is a measure of *linear* correlation between two series of data, is defined, for vectors of samples y and z , as

$$r_{ij} = \frac{\sum_{i=1}^N (y_i - y_{\text{mean}})(z_i - z_{\text{mean}})}{\sqrt{\sum_{i=1}^N (y_i - y_{\text{mean}})^2} \sqrt{\sum_{i=1}^N (z_i - z_{\text{mean}})^2}},$$

where y_{mean} and z_{mean} denote the sample means. Notice that the length N of the vectors y and z depends on the sampling time and on the window length. We argue that our metric is simply more convenient when dealing with periodic phase signals. While we could use the Pearson correlation coefficient to define functional pattern instead of our cosine-based metric, the former does not perform well on periodic signals that evolve on the unit circle, and is heavily dependent on the time window employed to collect the samples. Thus, specific

Figure 9: **Comparison between Pearson correlation coefficient and the metric $\rho_{12} = \langle \cos(\theta_2 - \theta_1) \rangle_t$ on two phase-locked signals and varying time window lengths.** Each point in the plot represents a value of ρ_{12} (in blue) and r_{12} (in red) computed in a time window $[0, k]$, where $k = 0.1, 0.2, \dots, 5$. It can be seen in all panels that ρ_{12} is only affected by the phase shift ψ . Instead, the Pearson correlation coefficient returns oscillating values for different window sizes with damping oscillations as the length of the time window increases. **a** The phase shift in the initial conditions $\psi = \frac{\pi}{8}$. **b** The phase shift in the initial conditions $\psi = \frac{\pi}{4}$. **c** The phase shift in the initial conditions $\psi = \frac{\pi}{3}$. **d** The phase shift in the initial conditions is $\psi = \frac{\pi}{2}$.

adjustments are needed to define correlation patterns through Pearson correlation coefficient for the class of time series (phase trajectories) studied in this work.

As an example, consider two identical sinusoidal signals (i.e., phase-locked) with natural frequency $\omega = 2\pi$ but shifted initial conditions: $\theta_1(0) = 0$, $\theta_2(0) = \varphi$, with $\varphi \in (0, \pi]$. Fig. 9 illustrates the differences between the Pearson correlation coefficient r_{12} and our correlation metric ρ_{12} computed over a time window of varying length and for different values of the initial phase shift φ . In all panels, the values of r_{12} vary at each point, emphasizing the dependence of the Pearson correlation coefficient from the length of the time window. Conversely, in all panels, the value of ρ_{12} remains unaltered by the choice of time window length.

In conclusion, we choose to define functional patterns through $\rho_{ij} = \langle \cos(\theta_j - \theta_i) \rangle_t$ because it is a convenient and suitable metric for this class of phase trajectories that naturally emerge in oscillator systems.

We have added this discussion in the Supplementary Information to better explain our choice of the correlation metric based on the cosine function.

[R2: 12] *The section “Assessing and enforcing stability of functional patterns” contains insufficient details to understand the proposed heuristic. It will be helpful if the steps can be illustrated.*

Response: Thank you for helping us improve the clarity of our manuscript. According to this reviewer’s suggestion, we have enhanced the description of the heuristic procedure and moved useful results from the

Supplementary Information to the Methods section. The text now reads:

“Note that the minimization problem (11) does not guarantee that the functional pattern \mathbf{x} is stable for the network with weights $\boldsymbol{\delta} + \boldsymbol{\alpha}^*$. To promote stability of the pattern \mathbf{x} , we use a heuristic procedure based on the classic Gerschgorin’s theorem [49]. Recall that stability of \mathbf{x} is guaranteed when the associated Jacobian matrix has a Laplacian structure, with negative diagonal entries and nonnegative off-diagonal entries. Further, instability of \mathbf{x} depends primarily on the negative off-diagonal entries $A_{ij} \cos(x_{ij})$ of the Jacobian (these entries are negative because the sign of the network weight A_{ij} is different from the sign of the cosine of the desired phase difference x_{ij}). Thus, reducing the magnitude of such entries A_{ij} heuristically moves the eigenvalues of the Jacobian towards the stable half of the complex plane (this phenomenon can be captured using the Gerschgorin circles, as we show in Fig. 10 for a network with 7 nodes). To formalize this procedure, let $\boldsymbol{\delta}_{\mathcal{N}}$ and $\boldsymbol{\alpha}_{\mathcal{N}}$ denote the entries of the weights $\boldsymbol{\delta}$ and tuning vector $\boldsymbol{\alpha}$, respectively, that are associated to negative interconnections $A_{ij} \cos(x_{ij}) < 0$ in the cosine-scaled network. Then, the optimization problem that enacts the proposed strategy becomes:

$$\begin{aligned} \min_{\boldsymbol{\alpha}} \quad & \|\boldsymbol{\delta}_{\mathcal{N}} + \boldsymbol{\alpha}_{\mathcal{N}}\|_2 \\ \text{subject to} \quad & BD(\mathbf{x})(\boldsymbol{\delta} + \boldsymbol{\alpha}) = \boldsymbol{\omega}, \\ & \text{and } (\boldsymbol{\delta} + \boldsymbol{\alpha}) \geq 0, \end{aligned} \tag{9}$$

Carefully reducing the weights $\boldsymbol{\delta}_{\mathcal{N}} + \boldsymbol{\alpha}_{\mathcal{N}}$ promotes stability of the target pattern. Fig. 10 illustrates the shift of the Jacobian’s eigenvalues while the optimal tuning vector $\boldsymbol{\alpha}^*$ is gradually applied to a 7-oscillator network to achieve stability of a functional pattern containing negative correlations (the network parameters can be found in the Supplementary Information). Finally, we remark that the procedure in (9) can be further refined by introducing scaling constants to penalize $\|\boldsymbol{\delta}_{\mathcal{N}} + \boldsymbol{\alpha}_{\mathcal{N}}\|_2$ differently from the modification of other interconnection weights (see Supplementary Information for further details and an example).”

Moreover, the Methods section contains now the following subsection:

“Heuristic procedure to promote stability of functional patterns in positive networks. We provide a heuristic procedure to promote the stability of functional patterns that include negative correlations in a network with nonnegative weights. Our procedure relies on the definition of Gerschgorin disks and the Gerschgorin Theorem.

Definition of Gerschgorin disk. Let $M \in \mathbb{C}^{n \times n}$ be a complex matrix. The i -th Gerschgorin disk is $\mathcal{D}_i = (M_{ii}, r_i)$, $i = 1, \dots, n$, where the radius is $r_i = \sum_{j \neq i} |M_{ij}|$ and the center is M_{ii} .

Theorem 2 (Gerschgorin [49]). *The eigenvalues of the matrix M lie within the union $\bigcup_{i=1}^n \mathcal{D}_i$ of its Gerschgorin disks.*

Whenever all target phase differences in \mathbf{x} satisfy $|x_{ij}| \leq \frac{\pi}{2}$, the Gerschgorin disks of the Jacobian in equation (10) all lie in the closed left half-plane. However, for patterns \mathbf{x} containing phase differences $|x_{ij}| \geq \frac{\pi}{2}$, the union of the Gerschgorin disks intersects the right half-plane. Reducing the magnitude of the entries satisfying $A_{ij} \cos(x_{ij}) < 0$ effectively shrinks the radius of the Gerschgorin disks that overlap with the right half-plane and shifts their centers towards the left-half plane due to the structure of the Jacobian matrix. We remark that the procedure in equation (9) is a heuristic, and it is provably effective only when all interconnections with $A_{ij} \cos(x_{ij}) < 0$ can be removed, so that all the Gerschgorin disks lie completely in the left-half plane.”

Finally, we added a figure in the main text that shows how the Gerschgorin disks and the Jacobian’s eigenvalues shift towards the (stable) left-half plane for the network in the example of Supplementary Text 1.5, which we display below.

Figure 10: **Mechanism underlying the heuristic procedure to promote stability of functional patterns containing negative correlations.** For the 7-oscillator network in Supplementary Text 1.5, we apply the procedure in equation (9) to achieve the stability of the pattern $\mathbf{x}_{\text{desired}} = [\frac{21\pi}{32} \frac{\pi}{6} \frac{\pi}{6} \frac{\pi}{8} \frac{\pi}{8} \frac{\pi}{3}]^T$, where $x_{12} = \theta_2 - \theta_1 > \frac{\pi}{2}$. The left plot illustrates the Gerschgorin disks (in blue) and the Jacobian's eigenvalues locations for the original network (as dark dots). The complex axis is highlighted in purple. It can be observed in the zoomed-in panel that one eigenvalue is unstable ($\lambda_2 = 0.0565$, in red). The optimal correction α^* is gradually applied to the existing interconnections from the left-most panel to the right-most one at $\frac{1}{3}$ increments. The right zoomed-in panel shows that, as a result of our procedure, $n - 1$ eigenvalues ultimately lie in the left-hand side of the complex plane ($\lambda_1 = 0$ due to rotational symmetry and $\lambda_2 = -0.0178$, in green).

BIBLIOGRAPHY

- [1] S. Chen, M. Fazlyab, M. Morari, G. J. Pappas, and V. M. Preciado. Learning region of attraction for nonlinear systems. *arXiv preprint arXiv:2110.00731*, 2021.
- [2] L. Zhu and D. J. Hill. Synchronization of Kuramoto oscillators: A regional stability framework. *IEEE Transactions on Automatic Control*, 65(12):5070–5082, 2020.
- [3] D. Mehta, N. S. Daleo, F. Dörfler, and J. D. Hauenstein. Algebraic geometrization of the Kuramoto model: Equilibria and stability analysis. *Chaos: An Interdisciplinary Journal of Nonlinear Science*, 25(5):053103, 2015.
- [4] F. Dörfler and F. Bullo. Synchronization in complex networks of phase oscillators: A survey. *Automatica*, 50(6):1539–1564, 2014.
- [5] A. Townsend, M. Stillman, and S. H. Strogatz. Dense networks that do not synchronize and sparse ones that do. *Chaos: An Interdisciplinary Journal of Nonlinear Science*, 30(8):083142, 2020.
- [6] G. S. Medvedev and X. Tang. Stability of twisted states in the Kuramoto model on Cayley and random graphs. *Journal of Nonlinear Science*, 25(6):1169–1208, 2015.
- [7] T. Menara, G. Baggio, D. S. Bassett, and F. Pasqualetti. Stability conditions for cluster synchronization in networks of heterogeneous Kuramoto oscillators. *IEEE Transactions on Control of Network Systems*, 7(1):302 – 314, 2020.
- [8] Juan A Acebrón, Luis L Bonilla, Conrad J Pérez Vicente, Félix Ritort, and Renato Spigler. The Kuramoto model: A simple paradigm for synchronization phenomena. *Reviews of modern physics*, 77(1):137, 2005.
- [9] J. Sun, E. M. Bollt, and T. Nishikawa. Master stability functions for coupled nearly identical dynamical systems. *EPL (Europhysics Letters)*, 85(6):60011, 2009.
- [10] H. K. Khalil. *Nonlinear Systems*. Prentice Hall, 3 edition, 2002.
- [11] K. Sharafutdinov, L. Rydin Gorjão, M. Matthiae, T. Faulwasser, and D. Witthaut. Rotor-angle versus voltage instability in the third-order model for synchronous generators. *Chaos: An Interdisciplinary Journal of Nonlinear Science*, 28(3):033117, 2018.
- [12] F. Hellmann, P. Schultz, P. Jaros, R. Levchenko, T. Kapitaniak, J. Kurths, and Y. Maistrenko. Network-induced multistability through lossy coupling and exotic solitary states. *Nature Communications*, 11(1):592, 2020.
- [13] Y. Susuki, I. Mezić, and T. Hikiyara. Coherent swing instability of power grids. *Journal of nonlinear science*, 21(3):403–439, 2011.
- [14] A. Moeini, I. Kamwa, P. Brunelle, and G. Sybille. Open data IEEE test systems implemented in SimPowerSystems for education and research in power grid dynamics and control. In *2015 50th International Universities Power Engineering Conference (UPEC)*, pages 1–6, 2015.
- [15] F. Dörfler and F. Bullo. Synchronization and transient stability in power networks and nonuniform Kuramoto oscillators. *SIAM Journal on Control and Optimization*, 50(3):1616–1642, 2012.

- [16] H. Sakaguchi and Y. Kuramoto. A soluble active rotator model showing phase transitions via mutual entrainment. *Progress of Theoretical Physics*, 76(3):576–581, 1986.
- [17] P. S. Skardal, D. Taylor, J. Sun, and A. Arenas. Collective frequency variation in network synchronization and reverse PageRank. *Physical Review E*, 93:042314, Apr 2016.
- [18] S. G. Krantz and H. R. Parks. *The Implicit Function Theorem: History, Theory, and Applications*. Birkhäuser, 2002.
- [19] T. Menara, G. Baggio, D. S. Bassett, and F. Pasqualetti. A framework to control functional connectivity in the human brain. In *IEEE Conf. on Decision and Control*, pages 4697–4704, Nice, France, December 2019.
- [20] M. Fazlyab, F. Dörfler, and V. M. Preciado. Optimal network design for synchronization of coupled oscillators. *Automatica*, 84:181–189, 2017.
- [21] F. Sorrentino, M. Di Bernardo, and F. Garofalo. Synchronizability and synchronization dynamics of weighed and unweighed scale free networks with degree mixing. *International Journal of Bifurcation and Chaos*, 17(07):2419–2434, 2007.
- [22] L. M. Pecora, F. Sorrentino, A. M. Hagerstrom, T. E. Murphy, and R. Roy. Cluster synchronization and isolated desynchronization in complex networks with symmetries. *Nature Communications*, 5(1):1–8, 2014.
- [23] István Z. Kiss, Craig G. Rusin, Hiroshi Kori, and John L. Hudson. Engineering complex dynamical structures: Sequential patterns and desynchronization. *Science*, 316(5833):1886–1889, 2007.
- [24] R. M. D’Souza and M. Mitzenmacher. Local cluster aggregation models of explosive percolation. *Physical Review Letters*, 104:195702, May 2010.
- [25] K. Bansal, J. O. Garcia, S. H. Tompson, T. Verstynen, J. M. Vettel, and S. F. Muldoon. Cognitive chimera states in human brain networks. *Science Advances*, 5(4), 2019.
- [26] Louis M. Pecora and Thomas L. Carroll. Master stability functions for synchronized coupled systems. *Physical Review Letters*, 80:2109–2112, Mar 1998.
- [27] S. H. Strogatz. *SYNC: The Emerging Science of Spontaneous Order*. Hyperion, 2003.
- [28] T. Nishikawa and A. E. Motter. Network synchronization landscape reveals compensatory structures, quantization, and the positive effect of negative interactions. *Proceedings of the National Academy of Sciences*, 107(23):10342–10347, 2010.
- [29] I. Belykh, R. Jeter, and V. Belykh. Foot force models of crowd dynamics on a wobbly bridge. *Science Advances*, 3(11):e1701512, 2017.
- [30] L. Papadopoulos, J. Z. Kim, J. Kurths, and D. S. Bassett. Development of structural correlations and synchronization from adaptive rewiring in networks of Kuramoto oscillators. *Chaos: An Interdisciplinary Journal of Nonlinear Science*, 27(7):073115, 2017.
- [31] A. Forrow, F. G. Woodhouse, and J. Dunkel. Functional control of network dynamics using designed Laplacian spectra. *Physical Review X*, 8(4):041043, 2018.
- [32] Y. Wang and F. J. Doyle. Exponential synchronization rate of Kuramoto oscillators in the presence of a pacemaker. *IEEE Transactions on Automatic Control*, 58(4):989–994, 2013.
- [33] D. J. Klein, P. Lee, K. A. Morgansen, and T. Javidi. Integration of communication and control using discrete time Kuramoto models for multivehicle coordination over broadcast networks. *IEEE Journal on Selected Areas in Communications*, 26(4):695–705, 2008.
- [34] T. Menara, G. Baggio, D. S. Bassett, and F. Pasqualetti. Conditions for feedback linearization of network systems. *IEEE Control Systems Letters*, 4(3):578–583, 2020.

- [35] S. P. Cornelius, W. L. Kath, and A. E. Motter. Realistic control of network dynamics. *Nature Communications*, 4, 2013.
- [36] C. Godsil and G. F. Royle. *Algebraic Graph Theory*. Graduate Texts in Mathematics. Springer New York, 2001.
- [37] G. Deco, V. Jirsa, A. R. McIntosh, O. Sporns, and R. Kötter. Key role of coupling, delay, and noise in resting brain fluctuations. *Proceedings of the National Academy of Sciences*, 106(25):10302–10307, 2009.
- [38] A. Ponce-Alvarez, G. Deco, P. Hagmann, G. L. Romani, D. Mantini, and M. Corbetta. Resting-state temporal synchronization networks emerge from connectivity topology and heterogeneity. *PLoS Computational Biology*, 11(2):1–23, 02 2015.
- [39] A. Politi and M. Rosenblum. Equivalence of phase-oscillator and integrate-and-fire models. *Physical Review E*, 91(4):042916, 2015.
- [40] J. Cabral, E. Hugues, O. Sporns, and G. Deco. Role of local network oscillations in resting-state functional connectivity. *NeuroImage*, 57(1):130–139, 2011.
- [41] F. Váša, M. Shanahan, P. J. Hellyer, G. Scott, J. Cabral, and R. Leech. Effects of lesions on synchrony and metastability in cortical networks. *Neuroimage*, 118:456–467, 2015.
- [42] P. Hövel, A. Viol, P. Loske, L. Merfort, and V. Vuksanović. Synchronization in functional networks of the human brain. *Journal of Nonlinear Science*, pages 1–24, 2018.
- [43] M. P. Van Den Heuvel and H. E. Hulshoff Pol. Exploring the brain network: a review on resting-state fmri functional connectivity. *European Neuropsychopharmacology*, 20(8):519–534, 2010.
- [44] F. C. Hoppensteadt and E. M. Izhikevich. *Weakly Connected Neural Networks*. Springer, 1997.
- [45] A. Daffertshofer and B. van Wijk. On the influence of amplitude on the connectivity between phases. *Frontiers in Neuroinformatics*, 5:6, 2011.
- [46] Joon-Young Moon, UnCheol Lee, Stefanie Blain-Moraes, and George A Mashour. General relationship of global topology, local dynamics, and directionality in large-scale brain networks. *PLOS Computational Biology*, 11(4):e1004225, 2015.
- [47] G. Deco, J. Cruzat, J. Cabral, E. Tagliazucchi, H. Laufs, N. K. Logothetis, and M. L. Kringelbach. Awakening: Predicting external stimulation to force transitions between different brain states. *Proceedings of the National Academy of Sciences*, 116(36):18088–18097, 2019.
- [48] A. Arenas, A. Díaz-Guilera, and C. J. Pérez-Vicente. Synchronization reveals topological scales in complex networks. *Physical Review Letters*, 96:114102, Mar 2006.
- [49] S. Geršgorin. Über die abgrenzung der eigenwerte einer matrix. *Bulletin de l'Académie des Sciences de l'URSS. Classe des sciences mathématiques et na*, pages 749–754, 1931.

REVIEWERS' COMMENTS

Reviewer #1 (Remarks to the Author):

In the first version of the manuscript, the authors have introduced a new method for functional control in systems of coupled heterogeneous Kuramoto oscillators. They have proposed a methodology, based on analytical existence criteria, that allows for finding structural configurations (either given by a certain weighted coupling structure and/or distribution of natural frequencies) such that the corresponding Kuramoto system possesses a prescribed phase-locked solution (functional pattern).

In the resubmitted manuscript, many new results are added that extend the scope of the proposed functional control method. In particular, a compelling discussion on the stability of the phase-locked states (functional patterns), including a Gerschgorin disks approach, and their coexistence (uniqueness) has been given. The authors show results on the basins of attraction as well as numerical evidence for the global stability of some of the patterns. I agree with the authors that estimating basins of attraction is a challenging problem that is far from solved. Therefore, I appreciate the author's efforts in providing new numerical results. These findings, in my opinion, are a strong supplement to the results presented in the original submission. Moreover, with their new results on the stability of phase-locked states, the authors provide even new insights on questions that relate to finite size systems of phase oscillators with distributed frequencies.

With regards to the generality of the functional control method, an extension to directed networks has been worked out. In the original submission, only symmetric coupling structures have been considered. The presented extension lifts this limitation of the original submission. A further generalization is provided by new results on the applicability of the proposed method for more realistic power grid models beyond the Kuramoto model.

In summary, all of my concerns have been addressed and I still believe that the proposed methodology will be of interest to researchers from various fields. In my opinion, the section "Contribution and future directions" provides an excellent overview of the work and describes many challenging future research directions. This renders the work to be potentially interesting to a very broad readership.

Minor comment:

- (i) In Fig. 7 of the main text: I suggest increasing the panel size of the blow-ups.
- (ii) With regards to the title, I agree with the authors and suggest also keeping the title as it is.

Reviewer #2 (Remarks to the Author):

Summary of content: In this manuscript, the authors investigate the problem of engineering a desired synchronization structure, referred to as Functional pattern, in a network of undirected Kuramoto oscillators by tuning the coupling strengths. To do this, they first establish sufficient conditions on the network topology and oscillator frequencies under which a desired functional pattern can be achieved by regulating the connection strengths. Second, they examine the conditions for the stability of functional patterns. Following this, they transform the task of constructing a desired stable functional pattern problem into a convex optimization problem in order to determine the optimal coupling weights or oscillation frequencies.

Resume: The presented study expands upon the idea introduced in [28] of tuning the connection weights to achieve a desired synchronization pattern in an oscillatory network. The type of problems as functional control has been extensively studied not only in control systems society but also in synchronization science and engineering. The techniques presented in this paper lack novelty. For instance, the optimization approach to pattern design and the Jacobian matrix-based method for stability analysis are standard techniques in optimal control theory and nonlinear dynamical systems theory. I would recommend to reject this work for publication in Nature Communications due to the following major technical shortcomings:

Page-3: The defined functional pattern $R=\rho_{ij}$, which the authors claim to be a metric, is not necessarily a positive-definite matrix and hence does not qualify as a metric.

Following definition, the introduced functional pattern matrix R is left untouched; only the phase differences are employed for pattern design and stability analysis. Moreover, phase differences do not correspond to the defined functional pattern in the one-to-one fashion due to the cosine being a non-injective function. In other words, synchronization patterns characterized by different pairwise phase differences might belong to the same functional pattern.

Even for a simple Kuramoto oscillator network, the reachability of a pattern by tuning the coupling strength depends on the initial phase of each oscillator, severely restricting the scope of the paper.

When many functional patterns exist, the authors make no attempt to analyze which pattern the network would eventually achieve. Moreover, is it always possible to assign one of the multiple feasible patterns to the network by adjusting the coupling strength?

Response to Reviewer 1

We thank the reviewer for their constructive comments and positive evaluation of our work.

- [R1: 1] *In the first version of the manuscript, the authors have introduced a new method for functional control in systems of coupled heterogeneous Kuramoto oscillators. They have proposed a methodology, based on analytical existence criteria, that allows for finding structural configurations (either given by a certain weighted coupling structure and/or distribution of natural frequencies) such that the corresponding Kuramoto system possesses a prescribed phase-locked solution (functional pattern). In the resubmitted manuscript, many new results are added that extend the scope of the proposed functional control method. In particular, a compelling discussion on the stability of the phase-locked states (functional patterns), including a Gerschgorin disks approach, and their coexistence (uniqueness) has been given. The authors show results on the basins of attraction as well as numerical evidence for the global stability of some of the patterns. I agree with the authors that estimating basins of attraction is a challenging problem that is far from solved. Therefore, I appreciate the author's efforts in providing new numerical results. These findings, in my opinion, are a strong supplement to the results presented in the original submission. Moreover, with their new results on the stability of phase-locked states, the authors provide even new insights on questions that relate to finite size systems of phase oscillators with distributed frequencies. With regards to the generality of the functional control method, an extension to directed networks has been worked out. In the original submission, only symmetric coupling structures have been considered. The presented extension lifts this limitation of the original submission. A further generalization is provided by new results on the applicability of the proposed method for more realistic power grid models beyond the Kuramoto model. In summary, all of my concerns have been addressed and I still believe that the proposed methodology will be of interest to researchers from various fields. In my opinion, the section "Contribution and future directions" provides an excellent overview of the work and describes many challenging future research directions. This renders the work to be potentially interesting to a very broad readership.*

Response: We thank the reviewer for the positive evaluation of our work and his/her thorough assessment of our manuscript.

- [R1: 2] *In Fig. 7 of the main text: I suggest increasing the panel size of the blow-ups.*

Response: Thank you for the suggestion. We have increased the panel size of the blow-ups.

- [R1: 3] *With regards to the title, I agree with the authors and suggest also keeping the title as it is.*

Response: Thank you.

Response to Reviewer 2

We thank the reviewer for their constructive comments.

[R2: 1] *Summary of content: In this manuscript, the authors investigate the problem of engineering a desired synchronization structure, referred to as Functional pattern, in a network of undirected Kuramoto oscillators by tuning the coupling strengths. To do this, they first establish sufficient conditions on the network topology and oscillator frequencies under which a desired functional pattern can be achieved by regulating the connection strengths. Second, they examine the conditions for the stability of functional patterns. Following this, they transform the task of constructing a desired stable functional pattern problem into a convex optimization problem in order to determine the optimal coupling weights or oscillation frequencies.*

Response: We thank the reviewer again for his/her comments on our work.

[R2: 2] *Resume: The presented study expands upon the idea introduced in [28] of tuning the connection weights to achieve a desired synchronization pattern in an oscillatory network. The type of problems as functional control has been extensively studied not only in control systems society but also in synchronization science and engineering. The techniques presented in this paper lack novelty.*

Response: We respectfully disagree. Following this reviewer's request in the previous review round, we have already highlighted the main differences from [28] both in the main text and in the previous *Statement of Changes and Revision*. The work in [28] is limited to prescribing an upper bound to the phase differences. In other words, the optimization there only allows to constrain the phase difference within a predefined arc length. From a technical standpoint, our work significantly departs from [28] by

- (i) enabling the prescription of precise *pairwise* phase differences (not bounds);
- (ii) investigating the stability properties of different functional patterns; and
- (iii) providing results on multistability of patterns.

We had also invited this reviewer in the previous *Statement of Changes and Revision* to provide any additional reference on functional control that he/she believes are related to this manuscript. We would be happy to compare our work with such references, but we haven't been able to find any despite our multiple searches.

[R2: 3] *For instance, the optimization approach to pattern design and the Jacobian matrix-based method for stability analysis are standard techniques in optimal control theory and nonlinear dynamical systems theory. I would recommend to reject this work for publication in Nature Communications due to the following major technical shortcomings*

Response: We thank the reviewer for this comment, but we fail to see how formulating a problem as an optimization problem automatically implies lack of novelty. Optimization is at the core of many engineering and scientific problems, and these are of interest to the readers of Nature Communications (see, for instance, Ref. [1, 2, 3, 4]). Further, the main contributions of this work do not lie in how we solve an optimization problem, but rather in providing an explicit solution to a novel synchronization problem (which to the best of our knowledge had not been considered before), in giving a qualitative and quantitative understanding of the feasibility and stability of functional patterns, and in applying these findings to a variety of applications.

[R2: 4] *Page-3: The defined functional pattern $R = \rho_{ij}$, which the authors claim to be a metric, is not necessarily a positive-definite matrix and hence does not qualify as a metric.*

Response: Thank you for pointing out this imprecision. We have corrected the main text and the Supplementary Information by removing the word “metric”. Following [5], we now introduce ρ_{ij} as a “local order parameter”.

[R2: 5] *Following definition, the introduced functional pattern matrix R is left untouched; only the phase differences are employed for pattern design and stability analysis.*

Response: The functional pattern R is a convenient way to graphically represent the pairwise, local correlations across the oscillators of a network. Similar correlation-based matrices are widely used, for instance, in neuroscience (where they are referred to as functional connectivity) to quantify the pairwise functional relationship between brain regions. For the derivations of our results, we use a simpler proxy to study and create functional patterns. Although the definition of functional pattern is not strictly necessary to derive the main results of the paper, it makes our framework and contributions more accessible and interpretable to a broad audience, and it leaves the door open to future studies that can consider the regulation of functional patterns in full generality.

[R2: 6] *Moreover, phase differences do not correspond to the defined functional pattern in the one-to-one fashion due to the cosine being a non-injective function. In other words, synchronization patterns characterized by different pairwise phase differences might belong to the same functional pattern.*

Response: The reviewer is correct that different sets of phase differences can generate the same functional pattern. However, this is not a limitation because the paper focuses on recreating a desired functional pattern and not in determining a unique set of phase differences associated with a functional pattern. Clearly, different choices of phase differences may have different stability properties, and the conditions derived in the paper can be used for this purpose.

In the revised manuscript we have clarified this aspect.

[R2: 7] *Even for a simple Kuramoto oscillator network, the reachability of a pattern by tuning the coupling strength depends on the initial phase of each oscillator, severely restricting the scope of the paper.*

Response: As for any nonlinear dynamical system, the convergence to a fixed point depends on the invariance of that specific point, its stability, and its region of attraction. In the manuscript we provide a comprehensive treatment of the aforementioned properties by studying invariance and stability in the main text, and addressing the estimation of regions of attraction in the Supplementary Information. Stability of functional patterns is typically local, as it is often the case for the equilibria of nonlinear systems. This is a feature of the complexity of the problem and it is not a limitation of our proposed solution, and certainly it does not limit the scope of this paper or any other paper ensuring local stability for nonlinear systems.

[R2: 8] *When many functional patterns exist, the authors make no attempt to analyze which pattern the network would eventually achieve.*

Response: The section *Stability of functional patterns* is in fact a first step to answer the Reviewer’s question, where we provide conditions to test for the stability and the instability of functional patterns. Predicting the functional pattern the network will converge to amounts to quantifying the basin of attraction of different equilibria in a nonlinear system. While numerical methods for this computation exist, this is an ongoing

research problem that is still far from seeing a complete analytical answer. Thus, the network converges to the functional pattern whose basin of attraction contains the oscillators initial phases. This can be computed or approximated numerically, but unfortunately no results exist at this point to obtain an analytical prediction.

[R2: 9] *Moreover, is it always possible to assign one of the multiple feasible patterns to the network by adjusting the coupling strength?*

Response: By adjusting the weights, one or multiple functional patterns become invariant and possibly stable and the network converges to one of them depending on its initial state. Further adjusting the coupling strengths may change the stability properties of the existing functional patterns or even change the set of feasible patterns. The conditions provided in the paper allow for the computation of the feasible patterns and provide an analysis of their stability properties as a direct function of the network weights and coupling.

BIBLIOGRAPHY

- [1] Z. Qin, B. G. Compton, J. A. Lewis, and M. J. Buehler. Structural optimization of 3d-printed synthetic spider webs for high strength. *Nature Communications*, 6(1):7038, 2015.
- [2] G. Naseri and M. A. G. Koffas. Application of combinatorial optimization strategies in synthetic biology. *Nature Communications*, 11(1):2446, 2020.
- [3] O. Borkowski, M. Koch, A. Zettor, A. Pandi, A. C. Batista, P. Soudier, and J. Faulon. Large scale active-learning-guided exploration for in vitro protein production optimization. *Nature Communications*, 11(1):1872, 2020.
- [4] P. Xu, Q. Gu, W. Wang, L. Wong, A. G. W. Bower, C. H. Collins, and M. A. G. Koffas. Modular optimization of multi-gene pathways for fatty acids production in e. coli. *Nature Communications*, 4(1):1409, 2013.
- [5] A. Arenas, A. Díaz-Guilera, and C. J. Pérez-Vicente. Synchronization reveals topological scales in complex networks. *Physical Review Letters*, 96:114102, Mar 2006.